# High-resolution land use and land cover dataset for regional climate modelling: Historical and future changes in Europe

Peter Hoffmann[1,2], Vanessa Reinhart[1,2], Diana Rechid[1], Nathalie de Noblet-Ducoudré[3], Edouard L. Davin[4,5,6], Christina Asmus[1,2], Benjamin Bechtel[7], Jürgen Böhner[8], Eleni Katragkou[9], and Sebastiaan Luyssaert[10]

[1]Helmholtz-Zentrum Hereon, Climate Service Center Germany (GERICS), Fischertwiete 1, 20095 Hamburg, Germany
[2]Universität Hamburg, Institute of Geography, Section Physical Geography, Bundesstraße 55, 20146 Hamburg, Germany
[3]Laboratoire des Sciences du Climat et de l'Environment, IPSL, Paris, France
[4]Wyss Academy for Nature, University of Bern, Bern, Switzerland
[5]Climate and Environmental Physics, Physics Institute, University of Bern, Bern, Switzerland
[6]Oeschger Centre for Climate Change Research, University of Bern, Bern, Switzerland
[7]Ruhr-Universität Bochum, Department of Geography, Universitätsstraße 150, 44801 Bochum, Germany
[8]Universität Hamburg, Institute of Geography, Cluster of Excellence "Climate, Climatic Change, and Society" (CLICCS), Bundesstraße 55, 20146, Hamburg, Germany
[9]Department of Meteorology and Climatology, School of Geology, Aristotle University of Thessaloniki, Greece
[10]Amsterdam Institute for Life and Environment, Vrije Universiteit Amsterdam, Amsterdam, 1081, The Netherlands

**Correspondence:** Peter Hoffmann (peter.hoffmann@hereon.de)

**Abstract.** Anthropogenic land-use and land cover change (LULCC) is a major driver of environmental changes. The biophysical impacts of these changes on the regional climate in Europe are currently extensively investigated within the WCRP CORDEX Flagship Pilot Study (FPS) LUCAS - "Land Use and Climate Across Scales" using an ensemble of different Regional Climate Models (RCMs) coupled with diverse Land Surface Models (LSMs). In order to investigate the impact of realistic LULCC on past and future climates, high-resolution datasets with observed LULCC and projected future LULCC scenarios are required as input for the RCM-LSM simulations. To account for these needs, we generated the LUCAS LUC dataset Version 1.1 at 0.1° resolution for Europe with annual LULC maps from 1950-2100 (Hoffmann et al., 2022b, a), which is tailored towards the use in state-of-the-art RCMs. The plant functional type distribution (PFT) for the year 2015 (i.e., LANDMATE PFT dataset) is derived from the European Space Agency Climate Change Initiative Land Cover (ESA-CCI LC) dataset. Details about the conversion method, cross-walking procedure and the evaluation of the LANDMATE PFT dataset are given in the companion paper by Reinhart et al. (2022b). Subsequently, we applied the land-use change information from the Land-Use Harmonization 2 (LUH2) dataset, provided at 0.25° resolution as input for CMIP6 experiments, to derive LULC distribution at high spatial resolution and at annual timesteps from 1950 to 2100. In order to convert land use and land management change information from LUH2 into changes in the PFT distribution, we developed a Land Use Translator (LUT) specific to the needs of RCMs. The annual PFT maps for Europe for the period 1950 to 2015 are derived from the historical LUH2 dataset by applying the LUT backward from 2015 to 1950. Historical changes in the forest type changes are considered using an additional European forest species dataset. The historical changes in the PFT distribution of LUCAS LUC follow closely the land use changes given by LUH2 but differ in some regions compared to other annual LULCC datasets. From 2016 onward, annual PFT



maps for future land use change scenarios based on LUH2 are derived for different Shared Socioeconomic Pathways (SSPs) and Representative Concentration Pathways (RCPs) combinations used in the framework of the Coupled Modelling Intercomparison Project Phase 6 (CMIP6). The resulting LULCC maps can be applied as land use forcing to the new generation of RCM simulations for downscaling of CMIP6 results. The newly developed LUT is transferable to other CORDEX regions world-wide.

# 1 Introduction

Human land surface modifications through land use are an important forcing on climate, and its direct biophysical effects on the local and regional climate can be as large as those associated with global greenhouse gas forcing (de Noblet-Ducoudré et al., 2012). Land use and land cover changes (LULCC) affect land-atmosphere processes through modifications of the surface energy balance (Mahmood et al., 2014; de Noblet-Ducoudré and Pitman, 2021). Up to now, LULCC forcing is not sufficiently accounted for in climate change projections conducted with regional climate models (RCMs) although the strongest impact of LULCC is found especially at those finer regional scales (Mahmood et al., 2014; Davin et al., 2014). Thus, robust fine-scale LULCC reconstructions are needed to quantify the interaction between regional and local biogeochemical and biophysical processes within RCMs, which may support effective land-use based climate adaptation as well as mitigation measures.

The first coordinated downscaling experiments including land use changes were performed in the frame of the WCRP CORDEX Flagship Pilot Study LUCAS Land Use and Climate Across Scales (LUCAS; Rechid et al., 2017). An ensemble of different RCMs coupled to diverse land surface models (LSMs) has been set up to perform idealized experiments with extreme LULCC scenarios for the EUR-44 domain driven by ERA-Interim reanalysis. The responses of the RCM-LSM ensemble to the two extreme LULCC scenarios show robust seasonal temperature signals for some regions and variables but disagreement for others between the different RCM-LSMs originating mainly from the different representation of land processes in the models (Davin et al., 2020; Breil et al., 2020).

In the next phases of LUCAS and within EURO-CORDEX (Jacob et al., 2020), it is planned to conduct simulations with past and future land-use changes on horizontal resolutions of ~12.5 km (i.e., EUR-11 domain) and for some specific sub-regions in Europe, simulations will be carried out with resolution down to convection permitting scales. This approach poses new requirements for LULCC reconstructions and scenarios:

1) A high spatial resolution (1 km or below) over an extent that covers the entire EURO-CORDEX domain. The high spatial resolution is needed to resolve regional features in a detail that enables investigating the impact of LULCC on small-scale processes such as local wind systems, convection, boundary layer processes, and scale-interactions (Mahmood et al., 2014) with RCMs.

2) A reconstruction going back 65 years in time including annual timesteps to perform attribution studies. Further, the LULCC product should extend 85 years into the future for analyzing the impact of several Shared Socioeconomic Pathways (SSPs) and Representative Concentration Pathways (RCPs) scenarios which could be used to improve projections of future regional climate change, accounting for both changes in human-induced atmospheric composition and land-cover and use.



3) LULCC forcing should necessarily follow the overall trends employed by the driving Global Climate Models/Earth System Models (GCM/ESM) to be consistent with the boundary forcing as it is done for other forcing data such as greenhouse gas concentrations or aerosol emissions (Taranu et al., 2022; Wohland, 2022).

4) Land use and land cover classes that are relevant for RCMs need to be included. At scales of ~50 km and lower, urban land cover plays a significant role (Chapman et al., 2019; Daniel et al., 2019; Katzfey et al., 2020). Moreover, at finer scales the ratio of needleleaf and broadleaf forest becomes a meaningful driver (Naudts et al., 2016; Schwaab et al., 2020). In addition, land management practices such as irrigation significantly alter local and regional climate (Lobell et al., 2009; Valmassoi et al., 2019) and should thus be accounted for in the reconstruction and scenarios.

The LULCC reconstructions applied within the Coupled Model Intercomparison Project Phase 6 (CMIP6; Eyring et al., 2016) and the Land Use Model Intercomparison Project (LUMIP; Lawrence et al., 2016) are harmonized with future projections based on SSP and RCP scenarios in order to generate the Land-Use Harmonization 2 dataset (LUH2; Hurtt et al., 2020). These land use and land management changes are available from 850 until 2100 (with extension until 2300) on a global 0.25° grid. Thus, LUH2 meets the requirement for the length of the dataset but not for spatial resolution. In addition, the land use classes

do not correspond to land use and land cover classes employed in most GCMs or RCMs.

Consequently, many modelling groups will have to convert the LUH2 land use changes into changes of land cover and land use input prior to their use in GCMs. For this conversion so-called Land Use Translators (LUTs) are applied (e.g., Di Vittorio et al., 2014; Mauritsen et al., 2019; Lurton et al., 2020), which are usually model specific. For most GCMs/ESMs, LUTs only account for changes in land use classes such as cropland, pasture and rangeland, and natural vegetation (e.g., Mauritsen et al.,

2019; Lurton et al., 2020). Within the natural vegetation class the relative distribution of vegetation types such as forest, shrubs or grassland is constant or computed by the dynamic vegetation model despite the fact that LUH2 provides information on changes in forested and non-forested vegetation. Keeping keeping the relative proportion of land-cover types constant in the natural fraction of the land in RCMs, which do not include dynamic vegetation models, would be a major limitation and would not meet the requirements listed above. In addition, urban changes are mostly discarded in LUT approaches because urban

land use is not considered by most GCMs, with some exceptions (e.g., Jackson et al., 2010; Danabasoglu et al., 2020; Katzfey et al., 2020). Consequently, we developed a new LUT approach, which also accounts for changes in the distribution of natural vegetation types and urban areas, and generated a new land cover input dataset for RCMs.

Inconsistencies due to coarse resolution of LUH2 are tackled to a large extent by applying the LUT to a high-resolution initial land cover dataset. There is a wide range of observed high-resolution land cover datasets available that have been used

to generate land cover input for RCMs (e.g., CORINE, MODIS, ESA-CCI LC, GlobCover, HILDA+). However, some of these datasets are only available for certain regions, such as CORINE (Jaffrain et al., 2017), which is only available for European Countries. Within CMIP6 GCMs/ESMs, the ESA Climate Change Initiative Land Cover product (ESA-CCI LC; ESA, 2017) is increasingly applied. The ESA-CCI provides a LC time series including annual maps from 1992 to 2018, at a global ~300 m grid, a resolution suitable for kilometer scale RCM simulations. The dataset shows good agreement with other land cover

products, globally and regionally (Achard et al., 2017; Reinhart et al., 2021). In addition, ESA-CCI LC has already been used for RCM studies on the impact of LULCC on the climate in Europe (Huang et al., 2020), where the potential for the use of





this dataset for RCMs was demonstrated. Reinhart et al. (2022b) developed a workflow to convert the ESA-CCI LC land cover classes into plant functional types (PFTs) as well as non-vegetated classes such as urban and bare ground for the European domain. The resulting LANDMATE PFT dataset (Version 1.1) (Reinhart et al., 2022a) shows a good agreement with ground truth observations and thus provides the initial land cover map for the LUT approach.


In this study, we introduce the new high-resolution LUCAS LUC historical/future land use and land cover change dataset (Version 1.1) (Hoffmann et al., 2022b, a), which we prepared to meet the requirements for the next generation RCM simulations for downscaling CMIP6 by the EURO-CORDEX community and in the framework of FPS LUCAS.

## 2 Methods and datasets

### 2.1 Workflow for generating the LUCAS LUC Dataset


The workflow to generate the LUCAS LUC dataset is shown in Fig. 1. It starts with the generation of a PFT map based on the ESA-CCI LC dataset, the so-called LANDMATE PFT dataset (Sect. 2.2.1). The methods and datasets used to create this dataset are described in the companion paper by Reinhart et al. (2022b). Therefore, only a short description of the basemap development is given in this paper. First, the ESA-CCI LC map for the year 2015, which has a native resolution of ~300 m globally, is aggregated to 0.1 ° resolution . Thereafter, the aggregated ESA-CCI LC map is converted into a set of PFTs. For the conversion of the ESA-CCI LC land cover classes into PFTs, a cross-walking procedure is commonly applied (Wilhelm et al., 2014; Li et al., 2018; Georgievski and Hagemann, 2019; Lurton et al., 2020; Reinhart et al., 2022b). Therefore, for each ESA-CCI LC land cover class we set-up a so-called cross-walking table (CWT), which defines the composition of this class in terms of PFT fractions. The CWTs are further refined based on climate zones, defined through the Holdridge Life Zones (HLZ; Wilhelm et al., 2014). The HLZ concept proposes a global classification of climatic zones in relation to potential vegetation cover dependent on mean annual precipitation data and mean monthly temperature (Holdridge, 1967). Supported by the HLZs, it is possible to customize the CWT for each ESA-CCI LC class in a way that fits to the respective climate region, which is of special importance when translating mixed vegetation classes into PFTs. The HLZs for the European domain are computed from atmospheric observations of temperature and precipitation taken from the E-OBS dataset and the CRU dataset (outside of E-OBS range). Using the higher resolved E-OBS dataset (0.1° resolution) instead of the rather coarse CRU dataset (0.5° resolution), as it was used by Wilhelm et al. (2014), allows for a more detailed representation of HLZs especially in regions with complex terrain. The distribution of C3 and C4 grasses within grassland areas, distinguished by some LSMs, is taken from a separate potential C4 map provided by the North American Carbon Program (NACP) for the Multi-scale Synthesis and Terrestrial Model Intercomparison Project (MsTMIP).


The information needed to compute changes in the PFT distribution is taken from the LUH2 dataset (Sect. 2.2.3), which provides land-use states and -transitions and land management information at a global 0.25° grid from 850 to 2015 (reconstructed) and from 2016 until 2100 (projected). The land-use classes and subsequently the land-use transitions mainly represent land used by humans through agriculture, livestock farming, forest management etc. and only distinguish between forested and non-



**Table 1.** Datasets employed in the present study.

| Dataset | Temporal Coverage | Spatial Coverage | Spatial Resolution | Reference |
|---|---|---|---|---|
| LUCAS LUC | | | | |
| LUCAS LUC historical Version 1.1 | 1950-2015 | Europe | 0.1° | Hoffmann et al. (2022b) |
| LUCAS LUC future Version 1.1 | 2016-2100 | Europe | 0.1° | Hoffmann et al. (2022a) |
| Employed for generating LUCAS LUC | | | | |
| LANDMATE PFT Version 1.1 | 2015 | Europe | 0.1° | Reinhart et al. (2022a, b) |
| LUH2-v2h, LUH2-v2h_high, LUH2-v2h_low | 850-2015 | global | 0.25° | Hurtt et al. (2020) |
| LUH2-v2f | 2016-2100 | global | 0.25° | Hurtt et al. (2020) |
| McGrath forest types | 1960-2010 | Europe | 0.5° | McGrath et al. (2015) |
| Employed for comparison | | | | |
| ESA POULTER PFTs from ESA-CCI LC | 1992-2018 | global | 0.1* | ESA (2017); Poulter et al. (2015) |
| MODIS PFTs from land cover collection (C6 MCD12Q1) | 2000-2018 | global | 500 m | Sulla-Menashe and Friedl (2018) |
| HILDA+ | 1950-2015 | global | 1000 m | Winkler et al. (2021) |

forested vegetation. Thus, they cannot directly be imposed onto the LANDMATE PFTs, which represent mainly the physical

land cover.

Instead, the LUH2 land-use transitions are translated into annual changes in PFT fractions for the historical period starting 2015 and going back until 1950 using the newly developed land use translator (LUT; Sect. 2.3). While LUH2 provides transitions of forest vegetation, historical changes in the forest type distributions are taken from a European forest area and species composition dataset provided by McGrath et al. (2015) (Sect. 2.2.2). By employing the LUT forward in time, the future annual

PFT changes are computed from the eight different land-use change scenarios provided by LUH2. In special cases, where a certain vegetation type is not present within a grid cell but should be increased according to the LUH2 and the rules provided by the LUT, a background map of potential vegetation is needed. This map is constructed from the ESA-CCI LC dataset as well as the CWT used for the LANDMATE PFTs (Sect. 2.2.1).

The final LUCAS LUC dataset consists of one file containing the annual PFT maps for the historical period from 1950 until

2015 (Hoffmann et al., 2022b) and eight different files for the land-use change scenarios for the future period from 2016 to 2100 (Hoffmann et al., 2022a). For the comparison of the historic land cover changes the ESA-CCI LC based PFT time series, the MODIS PFT dataset and HILDA+ are employed (Sect. 2.5). An overview over the datasets employed in this study is given in table 1.





**Figure 1.** Workflow for generating the LUCAS LUC dataset. The steps and datasets highlighted in the gray dashed box are described in detail by Reinhart et al. (2022b)

.





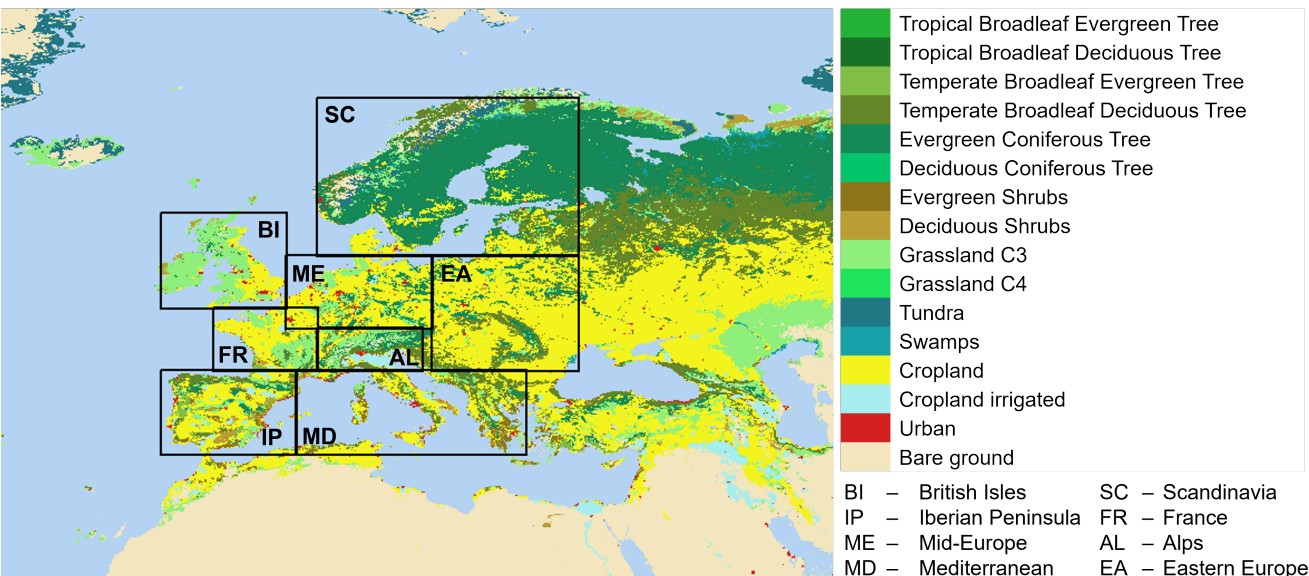

**Figure 2.** Distribution of the 16 LANDMATE PFTs at 0.1° resolution for 2015 based on ESA-CCI LC. Irrigation map from LUH2 is used to distinguish between irrigated crops and rainfed crops. For improved visualization the majority PFT is shown.

## 2.2 Land use and land cover datasets

### 2.2.1 LANDMATE PFT dataset Version 1.1

The LANDMATE PFT dataset Version 1.1 for the year 2015 is used as a basemap as a starting point for the land cover changes that are computed with the LUT (Sect. 2.3). Background maps, required for the LUT, are generated depending on the HLZ for the grass, shrub, and tree PFTs, respectively, based on the CWTs described in Reinhart et al. (2022b). For example, the background map for the tree PFT group consists of tree PFT fractions that are most likely to grow given the HLZ for each land point respectively. If applied to other regions, this map would need to be adjusted for the dominant vegetation cover of the region, e.g., for Australia, where temperate broadleaf evergreen forest is one of the dominant forest types.

The distribution of the PFTs in the LANDMATE PFT dataset (major PFT class per 0.1° grid cell) in 2015 is shown in Fig. 2. In many regions of Europe, cropland is the dominant PFT. In Scandinavia and Northern Russia, temperate evergreen forest is dominant, which changes into tundra at higher latitudes and altitudes. Even at a 0.1° resolution, urban land cover is the dominant land cover for a number of grid cells covering the major urban areas (e.g., London, Paris, Ruhr area etc.). This emphasizes the importance of including urban areas at resolutions employed in EURO-CORDEX (i.e., ~12.5 km and higher).



**Table 2.** LANDMATE plant functional types (PFTs) and non-vegetated classes based on Reinhart et al. (2022b). In addition, the grouping used within the LUT (Sect. 2.3) is given (i.e,. PFT group).

| Nr. | Names | PFT group |
|-----|-------|-----------|
| 1 | Tropical broadleaf evergreen trees | forest |
| 2 | Tropical deciduous trees | forest |
| 3 | Temperate broadleaf evergreen trees | forest |
| 4 | Temperate deciduous trees | forest |
| 5 | Evergreen coniferous trees | forest |
| 6 | Deciduous coniferous trees | forest |
| 7 | Coniferous shrubs | shrub |
| 8 | Deciduous shrubs | shrub |
| 9 | C3 grass | grass |
| 10 | C4 grass | grass |
| 11 | Tundra | grass |
| 12 | Swamp | no group |
| 13 | Non-irrigated crops | crop |
| 14 | Irrigated crops | crop |
| 15 | Urban | urban |
| 16 | Bare | no group |

#### 2.2.2 European forest area and species composition

The dataset from McGrath et al. (2015) provides tree species composition on a 0.5° grid for Europe from 1600 to 2010, taking into account the conversion of tree species due to forest management. The forest area was reconstructed based on the tree species maps of Brus et al. (2012), the land cover map of Poulter et al. (2015), and the historical land use maps of Kaplan et al. (2012, 2009). A detailed description of the tree species dataset is given by Naudts et al. (2016). While in Naudts et al. (2016) *Larix sp.* (the only deciduous coniferous species present in the dataset) is listed as a tree species, the fractions for this species are zero in the dataset. Consequently, no additional information for the time evolution is available for the deciduous coniferous PFT. The allocation of the tree species to the remaining three tree PFTs is provided in table 3.

#### 2.2.3 Land-use harmonized dataset version 2 (LUH2)

The Land-use harmonized dataset version 2 (LUH2; Hurtt et al., 2020) provides annual land-use states and transitions for 12 land use types (7 main land use types and 5 crop types, table 4) on a global 0.25° regular grid from 850 until 2100, with extension to 2300 (LUH2-v2h dataset for historic time period 850-2015 and LUH2-v2f dataset for future time period starting 2016). In addition, LUH2 provides also agricultural management information such as irrigation and fertilization. For the his-





**Table 3.** Allocation of the tree species data (McGrath et al., 2015; Naudts et al., 2016) to the tree PFTs. Please note that Naudts et al. (2016) distinguished the needleleaf species in temperate and boreal, even if they are the same species.

| PFT | tree species |
|---|---|
| Temperate broadleaf evergreen trees | *Quercus ilex and Q. suber* |
| Temperate deciduous trees | *Betula sp.*, *Fagus sylvatica*, *Quercus robur and Q. petraea*, *Populus sp.* |
| Evergreen coniferous trees | *Pinus sylvestris*, *Pinus pinaster*, *Picea sp.* |

torical period 850-2015 land-use changes are based on the History Database of the Global Environment version 3.2 (HYDE 3.2; Klein Goldewijk et al., 2017). In addition to the standard dataset, LUH2 provides two additional historical reconstructions (LUH2-v2h_high and LUH2-v2h_low) in order to provide uncertainty estimates for agricultural areas and wood harvesting taking from the HYDE3.2 dataset. For LUH2-v2h_high high historical estimates for crop and pasture and wood harvest compared to LUH2-v2h are assumed, whereas for LUH2-v2h_low low estimates are assumed (Lawrence et al., 2016; Hurtt et al.,

165 2020).

Future land-use changes are based on the output of different integrated assessment models (IAM) for selected marker SSP-RCP scenarios of which the most important characteristics are summarized in table A2. The Global Land Use Model (GLM2; Hurtt et al., 2006, 2011) is employed to translate the land-use change information into fractional changes of the land use classes for each 0.25° x 0.25° grid cell using additional datasets and assumptions as constrains. For three scenarios (i.e., SSP1/RCP1.9,

SSP1/RCP2.6, and SSP2/RCP4.5) an additional dataset is provided, which takes into account the future forestation that is present in these scenarios but was not captured in the initial LUH2 dataset (LUH2; Hurtt et al., 2020). The dataset contains the variable "added tree cover", which is the fraction of the 0.25° x 0.25° grid cell that should be converted from non-forested vegetation to forested vegetation to obtain the correct proportion of future forest cover.

### 2.3 Translating land use changes to PFT changes

In order to convert the land use change information given by LUH2 into PFT changes, an algorithm with a fixed set of transition rules is developed (Tables 5 and 6). In a first step LUH2 classes are grouped into main land use classes, denoted here as LUT classes (Table 4): Crops (CRO), forest (FOR), non-forest vegetation (NFV), rangeland (RAN), pasture (PAS), and urban (URB). For these LUT classes the transitions provided by the LUH2 dataset are aggregated. The aggregated transitions are bilinearly interpolated for the 0.25° grid to the 0.1° grid also used for the PFT maps derived from ESA-CCI LC (i.e. LANDMATE PFT

dataset; Sect. 2.2.1), which represent the land cover distribution for the year 2015.

The transition rules are defined to ensure that the changes in cropland are as close to the LUH2 changes as possible. In contrast to other LUTs, urban transitions are included in the LUT. Following the recommendations by Ma et al. (2020) and Hurtt et al. (2020), in the LUT natural vegetation (i.e., forest and shrubland) is only cleared and converted into grassland for land-use class transitions to pasture, while it remains unchanged for land-use class transitions from non-forested vegetation to



**Table 4.** Land cover classes used for the land use translator and their corresponding LUH2 land use classes.

| LUT classes | LUH2 classes |
| --- | --- |
| forest (FOR) | primary forest, secondary forest |
| non-forest vegetation (NFV) | primary non-forest, secondary non-forest |
| rangeland (RAN) | rangeland |
| pasture (PAS) | pasture |
| crops (CRO) | C3 annual crops, C3 perennial crops, C4 annual crops, C4 perennial crops, C3 nitrogen-fixing crops |
| urban (URB) | urban land |

rangeland. An exception to this general rule is the transition from forest to rangeland when the land will be used for livestock grazing.

The PFTs are increased or reduced according to the rules given in tables 5 and 6. The transitions are computed sequentially according to the numbering in tables 5 and 6. The forward translation starts with transitions from and to cropland followed by urban transitions. Thereafter, the remaining pasture and rangeland transitions are computed. Transitions from forest to non-

forested vegetation (i.e., shrubland and grassland) and vice versa are not considered in the forward translation because these fields are zero in original LUH2 scenario data. Consequently, future afforestation and deforestation only occur if land-use transitions related to land use classes urban, cropland, rangeland, and pasture are present. An exception is made for the three scenarios SSP1/RCP1.9, SSP1/RCP2.6, and SSP5/RCP4.5, where a separate dataset is provided for afforestation (Sect. 2.3.2). The backward translation also starts with cropland and urban transitions. Since the historical transitions from urban to any

other LUH2 land use class are zero, these transitions are not considered. The backward translation continues with the pasture and rangeland transitions.

Since the vegetation fractions differ between the LANDMATE PFT map, used as the basemap for LUCAS LUC, and LUH2 (e.g., the spatial distribution of forest fraction), the rules are designed to be flexible. In order to ensure that crop and urban changes are as close as possible to the changes provided by LUH2, transition to crops are not as strict regarding the treatment

of the PFTs that occupied the grid cell previously. For example, during the transition of forest to cropland (FOR2CRO in table 5) the LUT checks if enough tree PFTs are available. If this is not the case shrub PFTs are reduced and in a subsequent step also the grass PFTs, given that the sum of forest and shrub PFTs is still smaller than the transitions. The reduction of a PFT group is done until its fraction is zero.

For each transition in tables 5 and 6, the relative PFT-fractions remain constant within each PFT group (Table 2). For

example, an increase in non-forest vegetation would lead to an increase in all shrub and grass PFTs that are present within a grid cell. If a PFT class (e.g., tree PFTs) is not present in a certain grid cell but is supposed to increase, the relative fractions for this class are taken from the corresponding background map (Sect. 2.2.1). Bare ground and swamps remain unchanged because there is no information about bare ground or wetland changes in the LUH2 dataset and there is no additional information available that could justify a conversion of bare ground or wetlands to vegetation or crops or vice versa. Hence, land cover



changes related to desertification, cropland expansion into the desert, and drainage of wetlands are not included in the LUCAS LUC dataset.

### 2.3.1 Accounting for historical forest type distribution

For the backward extension of historical forest type distribution, additional information on the relative distribution of broad- and needleleaf forest taken from the McGrath dataset is employed (Sect. 2.2.2). To avoid the altering of the basemap derived from

ESA-CCI LC the relative fractions of three tree types (Temperate broadleaf evergreen, Temperate deciduous trees, Evergreen coniferous trees) are not directly imposed onto the PFT maps. Instead, only the trend in the relative fraction is used. For every timestep the difference in the relative fractions of the three PFTs are computed. These relative fraction changes are then converted into fraction changes of the individual PFTs by multiplying the relative fraction changes with the sum of the three PFTs.

### 2.3.2 Adding tree cover for future scenarios

For the three scenarios SSP1/RCP1.9, SSP1/RCP2.6, and SSP5/RCP4.5, respectively, an additional transition is computed because the afforestation signal is not correctly captured in the LUH2 land-use transitions (Hurtt et al., 2020). After the computation of transitions provided by LUH2 projections, the tree PFTs are increased by employing the added tree cover data (Sect. 2.2.3). Here, the same rules as for the transitions from forested vegetation to non-forested vegetation in the backward

translation are applied (FOR2NFV in table 6) increasing tree PFTs and reducing non-forested PFTs, starting with shrub PFTs and if their fraction is reduced to zero, grass PFTs.

### 2.3.3 Treatment of irrigated cropland

After the translation procedure, irrigated and non-irrigated crops are separated based on the irrigation fractions (e.g., irrigation of C3 annual crops) for the different crop classes provided by LUH2. These fractions are aggregated to create a single irrigation

fraction per grid cell. Within the irrigation fraction there is no consistent information on the irrigation practice (e.g., sprinkler or channel irrigation) available. After each transition timestep of one year the crop PFTs are summed up and multiplied with the irrigation fraction and (1 - irrigation fraction), respectively.

## 2.4 Uncertainty measures

To account for the main uncertainties of to the historical LULCC, two different historical reconstructions have been provided,

the so-called LUH2-v2h_low and LUH2-v2h_high dataset (Sect. 2.2.3). They are in turn based on the uncertainty estimates of the HYDE3.2 dataset for the population data and for the cropland and grazing land cover.

In order to quantify the uncertainty of the LUCAS LUC dataset for the historical period, the LUCAS LUC dataset has been generated using the three different LUH2 reconstructions (i.e. historical, historical high, and historical low) as well as generating the dataset at 0.1° resolution and at the native resolution of the LUH2 dataset of 0.25°. From these six datasets




**Table 5.** LUT rules for the translation of LUT class changes into into PFT changes forward in time using the PFT group definitions given in table 2. The order of the computations of transitions is given in brackets. Crops (CRO), forest (FOR), non-forest vegetation (NFV), rangeland (RAN), pasture (PAS), and urban (URB).

| | from FOR | from NFV | from CRO | from PAS | from RAN | from URB |
|---|---|---|---|---|---|---|
| to FOR | x | x | **CRO2FOR (5)** increase tree PFTs; reduce crop PFTs | **PAS2FOR (20)** increase tree PFTs; reduce grass PFTs | **RAN2FOR (23)** increase tree PFTs; reduce grass PFTs, if not available reduce shrub PFTs | **URB2FOR (16)** increase tree PFTs; reduce urban |
| to NFV | x | x | **CRO2NFV (6)** increase shrub & grass PFTs; reduce grass PFTs | **PAS2NFV (21)** increase shrub PFTs; reduce grass PFTs | x | **URB2NFV (15)** increase shrub & grass PFTs; reduce urban |
| to CRO | **FOR2CRO (1)** increase crop PFTs; reduce tree PFTs, if not available reduce shrubs PFTs, if not available reduce grass PFTs | **NFV2CRO (2)** increase crop PFTs; reduce shrubs PFTs, if not available reduce grass PFTs | x | **PAS2CRO (4)** increase crop PFTs; reduce grass PFTs | **RAN2CRO (3)** increase crop PFTs; reduce grass PFTs, if not available reduce shrub PFTs | **URB2CRO (14)** increase crop PFTs; reduce urban |
| to PAS | **FOR2PAS (18)** increase grass PFTs; reduce tree PFTs, if not available reduce shrub PFTs | **NFV2PAS (19)** increase grass PFTs; reduce shrub PFTs | **CRO2PAS (8)** increase grass PFTs; reduce crop PFTs | x | x | **URB2PAS (17)** increase grass PFTs; reduce urban |
| to RAN | **FOR2RAN (22)** increase grass PFTs; reduce tree PFTs | x | **CRO2RAN (7)** increase grass PFTs; reduce crop PFTs | x | x | **URB2RAN (16)** increase grass PFTs; reduce urban |
| to URB | **FOR2URB (10)** increase urban; reduce tree PFTs | **NFV2URB (11)** increase urban; reduce shrub PFTs, if not available reduce grass PFTs, if not available reduce tree PFTs | **CRO2URB (9)** increase urban; reduce crop PFTs | **PAS2URB (13)** increase urban; reduce grass PFTs | **RAN2URB (12)** increase urban; reduce grass PFTs, if not available reduce shrub PFTs | x |





**Table 6.** LUT rules for the translation of LUT class changes into PFT changes backward in time using the PFT group definitions given in table 2. Please note that the transitions provided by LUH2 are the same as in table 5 but the changes in PFTs given in this table are imposed backward in time. Crops (CRO), forest (FOR), non-forest vegetation (NFV), rangeland (RAN), pasture (PAS), and urban (URB).

|  | from FOR | from NFV | from CRO | from PAS | from RAN | from URB |
|---|---|---|---|---|---|---|
| to FOR | x | **NFV2FOR (22)** increase shrub and grass PFTs; reduce tree PFTs | **CRO2FOR (4)** increase crop PFTs; reduce tree PFTs, if not available reduce shrub PFTs | **PAS2FOR (15)** increase grass PFTs; reduce tree PFTs | **RAN2FOR (20)** increase grass & tree PFTs; reduce tree PFTs | x |
| to NFV | **FOR2NFV (21)** increase tree PFTs ; reduce shrub PFTs, if not available reduce grass PFTs | x | **CRO2NFV (2)** increase crop PFTs; reduce shrub PFTs, if not available reduce grass PFTs, if not available reduce tree PFTs | **PAS2NFV (18)** increase grass PFTs; reduce shrub PFTs | x | x |
| to CRO | **FOR2CRO (3)** increase tree PFTs; reduce crop PFTs | **NFV2CRO (1)** increase shrub PFTs; reduce crop PFTs | x | **PAS2CRO (7)** increase grass PFTs; reduce crop PFTs | **RAN2CRO (5)** increase grass PFTs; reduce crop PFTs | x |
| to PAS | **FOR2PAS (14)** increase tree PFTs; reduce grass PFTs | **NFV2PAS (16)** increase shrub PFTs; reduce grass PFTs | **CRO2PAS (8)** increase crop PFTs; reduce grass PFTs | x | **RAN2PAS (17)** increase shrub PFTs; reduce grass PFTs | x |
| to RAN | **FOR2RAN (19)** increase tree PFTs; reduce shrub PFTs, if not available reduce grass PFTs | x | **CRO2RAN (6)** increase crop PFTs; reduce grass PFTs | x | x | x |
| to URB | **FOR2URB (11)** increase tree PFTs; reduce urban | **NFV2URB (10)** increase shrub PFTs; reduce urban | **CRO2URB (9)** increase crop PFTs; reduce urban | **PAS2URB (13)** increase grass PFTs; reduce urban | **RAN2URB (12)** increase grass and shrub PFTs; reduce urban | x |

(three reconstructions per resolution) two measures were derived. Following Winkler et al. (2021) the uncertainty is defined as the spread of a given PFT in a grid cell. Consequently, changes can be defined as robust if their absolute values are larger than the spread. As an aggregated measure of robustness the fraction of robust changes for a given PFT or land cover category and a given region, i.e. the ratio of robust changes to all changes. For the computations only fraction changes of 0.01 (i.e.,1% of the grid cell) and larger are considered.





The second measure is the agreement in the sign of the change, which is a widely used measure for the robustness of changes in climate research. For each of the six datasets the changes in PFTs are computed with respect to their basemap for the year 2015. The measure is 1 if all of the datasets show the same direction of changes (i.e., all decrease or all increase) and 0 otherwise. It is used for generating the historical land cover change maps (Figures 3 and 6) in this paper.

## 2.5    Comparison with existing LULCC datasets

In addition to the uncertainty analysis, the historical trends of LUCAS LUC PFTs are compared to the trends in the ESA-CCI based PFT time series (i.e., ESA POULTER, Sect. 2.5.1), the MODIS PFT time series (Sect. 2.5.2), and the land use change dataset HILDA+ (Sect. 2.5.3). The land use states (LUH2), land use classes (HILDA+), and PFTs (MODIS, LUCAS LUC, ESA POULTER) are aggregated to the land cover groups cropland, grassland, forest, and urban according to table 7. Note that for this purpose the LUH2 land use class pasture is assigned to grassland because LUH2 does not provide a grassland type and that

only the classes urban, cropland, and forest are taken from HILDA+ because no distinction between shrubs and grassland was made for this dataset. The spatial extent of the land cover fraction changes (Fig. 3) between 1992 and 2015, the period covered by ESA POULTER, as well as the time series of aggregated area changes (Fig. 4) are investigated. For the latter analysis, land cover fractions are converted into area coverage per year for eight European sub-regions defined within the project Prediction of Regional scenarios and Uncertainties for Defining European Climate change risks and Effects (PRUDENCE; Christensen

et al., 2007; Christensen and Christensen, 2007, Fig. 2) taking into account the land-sea mask from ESA POULTER, which is the same for LUCAS LUC. Results will be shown for the IP - Iberian Peninsula, ME - Mid-Europe, and EA - Eastern Europe and discussed as these regions are representative to illustrate the strengths and weaknesses of the high-resolution LULCC reconstruction (Sect. 3.1).

### 2.5.1    Historical PFT time series based on ESA-CCI LC PFTs

Together with the high-resolution land cover maps, the ESA-CCI provides a dedicated user tool to re-project and re-sample the LULC maps and to translate the LULC classes into model specific PFTs. During the re-sampling from the native ~300 m horizontal resolution, the LULC class fractions are automatically preserved as fractions per re-sampled grid cell. The user tool provides a generic translation table but also gives the possibility to include user-defined translations. Further, the involvement of climate data within the translation process is possible to a limited extent. In order to prepare the ESA-CCI-based PFT maps

for the present comparison, the generic table provided by ESA is used under consideration of the modifications introduced by Poulter et al. (2015). In addition to adjustments of the LULC class translation, an urban-PFT is added (Table A4). The resolution of the aggregated PFT maps can be chosen flexibly as required by the user. For the comparison with the LUCAS LUC PFT maps, the PFT maps, denoted as ESA POULTER in the following, are aggregated to a horizontal resolution of 0.1°.



**Table 7.** Harmonized land cover groups and the corresponding PFTs and land use classes from LUCAS LUC, MODIS, ESA POULTER, HILDA+, and LUH2.

| land cover groups | LUCAS LUC PFTs | MODIS PFTs | ESA POULTER PFTs | HILDA+ Class | LUH2 Class |
|---|---|---|---|---|---|
| cropland | Non-irrigated crops, irrigated crops | Cereal Croplands, Broadleaf Croplands | crop | Cropland | C3 annual crops, C3 perennial crops, C4 annual crops, C4 perennial crops, C3 nitrogen-fixing crops |
| grassland | C3 grass, C4 grass | grass | natural grass | pasture, Grass/Shrubland | pasture |
| forest | Tropical broadleaf evergreen trees, Tropical deciduous trees, Temperate broadleaf evergreen trees, Temperate deciduous trees, Evergreen coniferous trees, Deciduous coniferous tree | Evergreen Needleleaf Trees, Evergreen Broadleaf Trees, Deciduous Needleleaf Trees, Deciduous Broadleaf Trees | broadleaf evergreen, broadleaf deciduous, needleleaf evergreen, needleleaf deciduous | forest | primary forest, secondary forest |
| urban | urban | Urban and Built-up Lands | urban | urban | urban land |

### 2.5.2 Historical PFT time series based on MODIS

The Collection 6 Terra and Aqua combined Moderate Resolution Imaging Spectroradiometer (MODIS) land cover datasets (C6 MCD12Q1) provide ready to use PFT maps as one of their 13 science data sets. For C6 MCD12Q1, several processing steps were refined to eliminate known issues from the MODIS Collection 5 data sets, such as the excessively high inter-annual variability (Abercrombie and Friedl, 2015). Additional information on the MODIS data processing can be found at https://lpdaac.usgs.gov/products/mcd12q1v006/. The annual maps are available globally from 2001-2018 in ~500 m horizon-

tal resolution. The 12 MODIS PFTs follow the PFT definition that was developed for the National Center for Atmospheric Research land surface model (NCAR LSM) (Bonan et al., 2002). Table A3 shows the 12 PFTs including their description. The 13 science datasets provided within C6 MCD12Q1 are available in six different LULC classifications, including the PFT classification. All LULC classification are generated through employment of a supervised classification algorithm (Sulla-Menashe





and Friedl, 2018). For the comparison with the LUCAS LUC dataset, the MODIS PFT maps are aggregated to 0.1° horizontal
resolution.

### 2.5.3  Historical land use and land cover time series based on HILDA+

The HIstoric Land Dynamics Assessment + (HILDA+; Winkler et al., 2021) dataset provides global land use change informa-
tion for land use/cover classes (urban, cropland, pasture/rangeland, forest, unmanaged grass/shrubland, sparse/no vegetation) at
1 km resolution from 1950 to 2019. The basemap was generated from the Copernicus LC100 dataset Tsendbazar et al. (2021).
The land use changes are taken from multiple global (e.g., MODIS MCD12Q1 and ESA-CCI LC) and regional (e.g., CORINE)
sources.

## 3  Results

### 3.1  Historical LULC

#### 3.1.1  Cropland

Cropland changes in LUCAS LUC correspond well with the changes in LUH2, with some exceptions (Fig. 3a,b). The increase
in cropland in Iceland, visible in LUH2, is not present in LUCAS LUC because the LANDMATE PFTs, which is used as
basemap for LUCAS LUC, has no cropland in Iceland in 2015 (not shown). Moreover, weaker changes in cropland fractions
in LUCAS LUC compared to LUH2 are found in the Middle East and in Northern Africa. The LUT keeps the fraction of
bare ground constant (Sect. 2.3), which limits the magnitude of possible land cover changes in regions with large bare ground
fractions.

The decrease in cropland fraction in parts of Eastern Europe is present in both HILDA+ and ESA POULTER, while the latter
one shows a smaller magnitude than LUCAS LUC and LUH2, respectively (Fig. 3c,d). The cropland reduction in Central- and
Southern Europe in LUCAS LUC is also visible in ESA POULTER. In contrast, HILDA+ shows cropland increases for large
regions in Spain, France, and Germany. The strong increase in cropland in Southern Russia and Northwestern Kazakhstan
found in ESA POULTER and HILDA+ is not captured by LUCAS LUC and LUH2. In Southern Scandinavia and Estonia, the
cropland signals in LUCAS LUC, LUH2, and HILDA+ are opposite in comparison to the ESA POULTER derived signal. For
Egypt, ESA POULTER and HILDA+ show a reduction of cropland for the Central Nile Delta and along the Nile River and an
increase along the Edges of the Nile Delta whereas LUCAS LUC shows only an increase in this region. The increase along the
delta edges can be attributed to cropland expansion and the decreases to urbanization (Xu et al., 2017). These small-scale land
cover dynamics are not captured by LUH2.

The decreasing trend in cropland is visible in the aggregated values for all datasets in the three PRUDENCE regions EA,
ME, and IP (Fig. 4a-c). Only in MODIS, some years show less cropland compared to the year 2015. LUCAS LUC follows
the LUH2 annual area changes closely but with a slightly lower magnitude. MODIS and ESA POULTER show much smaller
changes in EA and IP, while ESA POULTER surpasses the LUCAS LUC and LUH2 changes in the period 1992 to 2001 in





ME. HILDA+ shows smaller changes in ME than LUCAS LUC. For IP, LUCAS LUC and HILDA+ are very close before 1990 and deviate thereafter. The overall spread of the different LUCAS LUC time series increase with time, originating mainly from the different LUH2 reconstructions. This uncertainty is small compared to the actual changes.

In LUCAS LUC, cropland decreases are even more widespread for most parts of Europe when starting from 1950 (Fig. 6a) instead of 1992 (Fig. 3a). Especially, in Mid-Europe a steep decline in cropland cover is visible from the 1950s to the
1970s (Fig. 4b). On the other hand, increase in Northern Africa, Iran, and along the Nile Delta are larger for the longer time period. In addition, the areas with increasing cropland cover in Southern Russia and Eastern Ukraine expanded compared to the 1992-2015 period. The largest spread in the changes signals can be found in parts of Italy, Poland, and Denmark (Fig. 6b). Overall, the fraction of robust changes is 98.9% for Europe but varies between 95.6% in AL and 99.9% in BI (Table 8).

### 3.1.2 Irrigated cropland

As described in sect. 2.3.3, the area development of irrigated cropland follows the trend of LUH2. From 1950 to 2015, the fraction of irrigated cropland increases in most countries in the research area (Fig. 6c). In the western Mediterranean regions, this increase is caused by the decrease of cropland, whereas in the Balkan regions, in the Middle East as well as in the Transcaucasian region the relative fraction of irrigated cropland increases. The increase is predominantly evident along freshwater sources such as rivers and channels, lakes as well as aquifers. In particular, the increase in irrigated cropland is striking along
the Garonne in France, along the Ebro in Spain, along the Euphrates and Tigris in Iraq as well as along the Nile in Egypt. Following the results of Thebo et al. (2014), irrigated cropland also appears increasingly around cities (e.g., around Paris in France, Casablanca in Morocco, Tripoli in Libya). The spread of the climate change signal is large in many parts of the Mediterranean region and the Middle East, with the largest spread in Iraq (Fig. 6d). However, the spread is still smaller than the changes resulting in fractions of robust changes above 95% for all regions (Table 8).

### 3.1.3 Grassland

While LUH2 shows a decrease in grassland (i.e. land use class pasture) in Spain and Poland (Fig. 3f), LUCAS LUC shows an increase in grassland (Fig. 3e). The reduction in grassland in LUH2 is mainly driven by conversion of non-forested vegetation to pasture, which compensates for the increase in pasture converted from cropland. However, in LUCAS LUC the non-forested vegetation that can be converted into grassland is shrubland (Tables 5 and 6), which is sparsely present in Poland and the parts
of Central Spain (not shown). This limits to what extent the LUT can increase grassland by decreasing shrubs. Consequently, other transitions (e.g., from cropland to pasture) dominate the land cover change signal for grassland. While LUCAS LUC grassland changes do not follow the pasture changes in LUH2 they are closer to the strong pasture increase in Poland found by Kuemmerle et al. (2016), who used data from the Common Agricultural Policy Regionalized Impact (CAPRI) database. Just like the cropland changes, alterations of grassland fractions are weaker in LUCAS LUC compared to the pasture changes in
LUH2 in the Middle East (Fig. 3e,f). This can be partly attributed to the larger share of bare ground in this region in LUCAS LUC.

The changes in grassland are quite small in ESA POULTER except for Southern Russia and Northwestern Kazakhstan, where a dual pattern of decrease and increase can be seen (Fig. 3g). The decrease in grassland in this region results from a conversion into cropland (Fig. 3c), which LUH2 does not capture.

The time series of grassland changes for the three PRUDENCE regions show substantial differences between the datasets (Fig. 4d-f). Since LUH2 does not provide grassland cover, pasture area changes are plotted instead. This is likely the reason why LUCAS LUC changes divert more strongly from LUH2 when changes in grassland are compared. For IP and EA, the spread of the change signal is small compared to the magnitude of the change signal, while it is of similar size in ME. In contrast to the cropland changes, the spread originates mainly from the difference in the resolution between LUCAS LUC and

LUH2.

The grassland cover increases in many of the east European countries from 1950 to 2015 with the exception of Estonia, Slovenia, and the Czech Republic (Fig. 6e), which show a decrease in grassland. In addition, Portugal and Turkey show increase in grassland. Decreasing grassland cover is found in Central and Southern Europe as well as in Scotland and parts of Russia. In contrast to the shorter period from 1992 to 2015, the grassland cover increases in Northern Germany, Northern

France, England, and Ireland between 1950 and 2015. The grassland changes show a spread throughout most of Southern-, Central-, and Eastern Europe (Fig. 6f). This also results in a lower fraction of robust changes in comparison with the other land cover classes (Table 8). For the European domain, the fraction is 94.4% but ME, FR, and BI have only a fraction of 82.5% to 86.4%.

### 3.1.4 Forest

Forest changes in LUCAS LUC match the LUH2 changes closely (Fig. 3h,i). Increases in forest fractions in mountainous areas (e.g., Alps, Carpathians Mountains, Balkan Mountains, and Pyrenees), Great Britain, Ireland, and in Russia between 55° and 60° are found in both datasets. The increases in Northern Italy and Ireland are weaker in LUCAS LUC compared to LUH2. This is again caused by the difference in forest cover in LUCAS LUC and LUH2 in the year 2015. The LUT can only decrease as much forest fraction backward in time (i.e., increase for the comparison between 1992 and 2015) as there is available in

2015. Decreasing forest fraction can be seen in Russia (e.g., east of Belarus and at the border to Georgia) and in Albania. As for cropland and grassland, forest changes in LUH2 and thus also in LUCAS LUC differ in many regions compared to ESA POULTER and HILDA+ (Fig. 3j,k). The change signals tend to be reversed between LUCAS LUC/LUH2 and ESA POULTER, with a decrease in mountainous areas (e.g., Alps, Pyrenees, and Balkan Mountains) and a increase near the Russian border to Belarus. Especially the decrease in the Alps and Pyrenees seem to be not supported by HILDA+ and recent assessments. For

example, Fernández-Nogueira and Corbelle-Rico (2018) found a strong afforestation signal between 1990 and 2012 in the northern parts of Spain based on the CORINE Land Cover dataset. In addition, inventory based datasets show an increase in forest cover during the 1990s and the 2000s for the Alps (e.g., Schwaab et al., 2015; Bebi et al., 2017). Substantial forest fraction increases in Northern Russia, Northern Norway, and Northern Finland can be seen in ESA POULTER and HILDA+. In these regions, both LUCAS LUC and LUH2 show no change in forest fraction at all. In these areas forest cover increased





due to forest growth instead of active afforestation (Potapov et al., 2015), which seems to be captured by the satellite-based ESA POULTER dataset but not by LUH2.

    Forest cover increases in both LUH2 and LUCAS LUC in the PRUDENCE regions EA, ME, and MD (Fig. 4g-i). In EA, ESA POULTER also shows a increase in forest cover but with a smaller magnitude while MODIS forest cover shows both increase and decreases. The difference between LUH2 and LUCAS LUC compared to the two satellite-based datasets are more

substantial in ME. Here, ESA POULTER shows a strong decrease while MODIS shows both strong increases and decreases for some years. In the IP region, LUCAS LUC, LUH2, MODIS, and in especially HILDA+ shows an increase in forest cover while ESA POULTER shows a decrease. The increases are much more pronounced in the HILDA+ dataset. The afforestation trend in IP is mostly due to farmland abandoning (Vilà-Cabrera et al., 2017; Palmero-Iniesta et al., 2021). While the cropland decrease is also present in LUCAS LUC cropland is mainly converted into shrubland except for Northern Spain (not shown).

The expansion in forest cover in LUCAS LUC is more pronounced for the period 1950 to 2015 (Fig. 6g) than for the period 1992 to 2015 (Fig. 3). Especially in the Alps, Balkan, Caucasus, Scotland, Estonia, and Lithuania forest cover increases substantially. In addition, for the longer time period forest cover increase is found in Sweden and Finland. An increase in forest cover in Finland was also found by Gao et al. (2014), who related these changes to the conversion of peatland into forest. Areas with forest cover reduction in Russia are smaller in extent and in magnitude compared to the period 1992 to 2015. Interestingly,

Iceland experienced a decrease in forest over the longer time period compared to an increase for the shorter time period, which is likely due to the strong government-lead reforestation efforts initiated in the 1980s and 1990s (Halldorsson et al., 2008). The spread of the LUCAS LUC reconstructions is noticeable in regions with a strong forest cover increase (Fig. 6h). For Europe, 95.4% of changes are widely robust, only BI and EA have a lower fraction of robust changes. In addition, the extent of slight increases is uncertain while lager changes in this region are more robust (not shown).

**3.1.5  Urban**

Changes in urban areas are limited to the major urban agglomerations in LUCAS LUC and LUH2 (Fig. 3l,m) while ESA POULTER (Fig. 3n) shows a more widespread urbanization also in rural areas in Central and Eastern Europe. HILDA+ shows widespread increases but also decreases in some parts of Central and Eastern Europe (Fig. 3o). Aggregated for the EA, ME, and IP regions all datasets show an urbanization signal (Fig. 4j-l). LUCAS LUC and HIDLA+ show a larger trend in the

European urbanization signal before 1990, which was also found in other studies (e.g., Güneralp et al., 2020; Tian et al., 2022). In contrast, the rapid urbanization in the ESA POULTER dataset in the early 2000s seems to be too strong. Reinhart et al. (2021) showed that the urban area fraction in ESA-CCI LC increased by 60% between 2000 and 2006 in eastern European countries compared to 6% in CORINE. Overall the different land cover datasets show large differences in the urbanization trend, with LUH2 and, therefore, LUCAS LUC at the moderate end and HILDA+ at the upper end.

The urban fraction in LUCAS LUC increase for the period 1950 to 2015 (Fig. 6i) is larger and more wide-spread compared to the signal from 1992 to 2015 (Fig. 3l). A strong urbanization signal is visible for Madrid, Paris, and Moscow. A larger scale increase is found in England, Northern Italy, Benelux region, Western Germany, Poland, and Slovenia. The urbanization signal is hardly affected by the uncertainties due to the LUH2 reconstructions but shows smaller changes in the 0.1° dataset compared



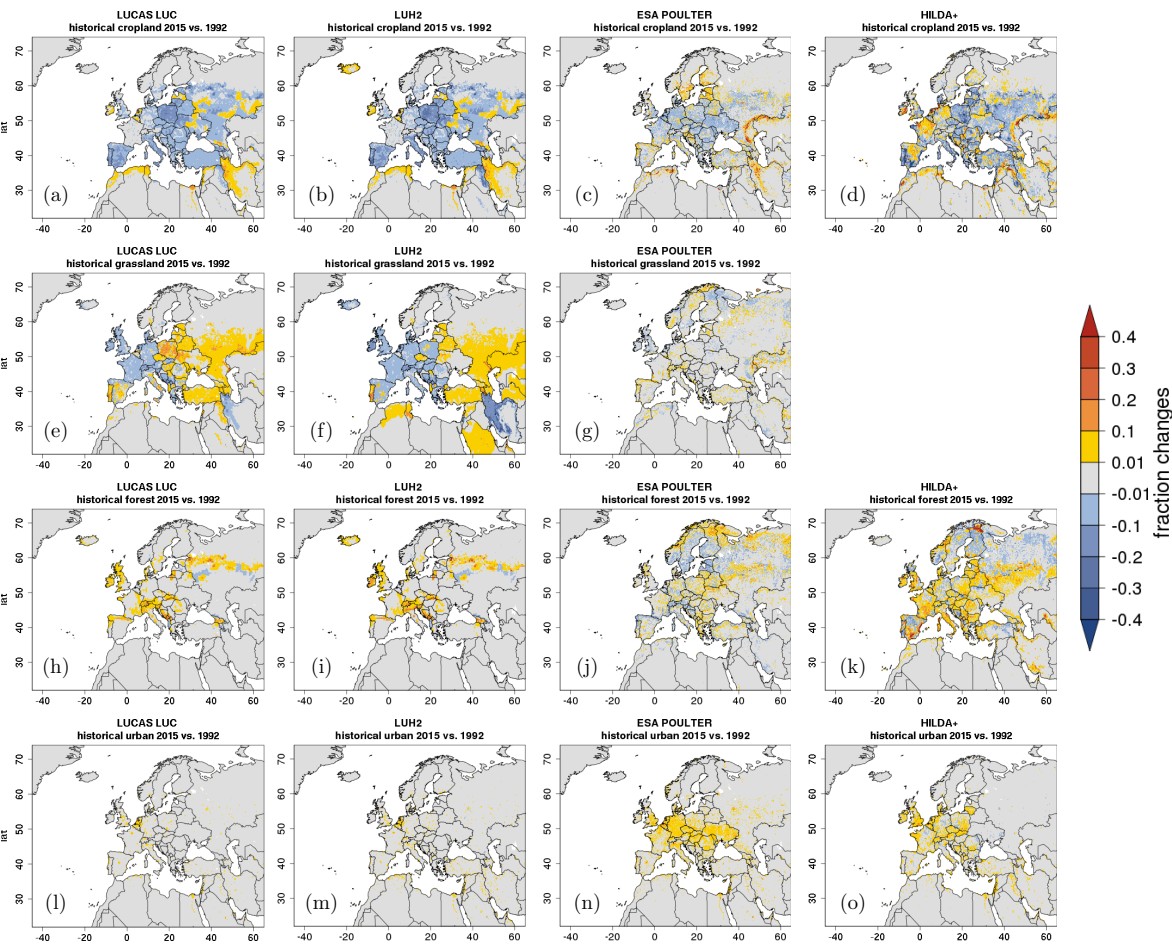

**Figure 3.** Changes in grid cell fraction for cropland (a-d), grassland (e-g), forest (h-k), and urban (l-o) classes based on LUCAS LUC (a,e,h,l), LUH2 (b,e,k,h), ESA POULTER (c,f,i,l), HILDA+ (d,k,o) between 1992 and 2015. For LUCAS LUC only the changes are show, where all LUCAS LUC reconstructions agree in the sign of the change (Sect. 2.4). Please note that for LUH2 pasture is taken as the grassland class and HILDA+ does not provide a grassland class.

to the 0.25° dataset, which is closer to the LUH2 changes (Fig. Fig. 4j-l). Despite these differences, the spread is smaller than

the actual changes, which leads to a high number of robust changes in all regions (Table 8).

### 3.1.6 Forest type

In Fig. 5, the annually averaged broad-/needleleaf forest ratio of the LUCAS LUC dataset with and without employing McGrath data (Sect. 2.3), the original McGrath dataset, ESA POULTER, and MODIS for different PRUDENCE regions is presented. Without employing the McGrath dataset the broad-/needleleaf ratio is almost constant throughout the historical period because

the relative fractions within a PFT group are preserved during the LUT transition computation. As intended, the trends in

**Figure 4.** Area changes with respect to the year 2015 in cropland (a-c), grassland (d-f), forest (g-i), and urban (j-l) computed for LUCAS LUC, LUH2, ESA POULTER, MODIS, and HILDA+ for the PRUDENCE regions Iberian Peninsula (a,d,g,j), Mid-Europe (b,e,h,k), Eastern Europe (c,f,i,l). In addition, values for LUCAS LUC at different resolution and based on different LUH2 reconstruction are provided. Please note that for LUH2 pasture is taken as the grassland class.





**Table 8.** Percentage of robust changes (Sect. 2.4) with an absolute value > 0.01 for the different land cover types for Europe (25°W-50°E, 30°N-70°N) and the PRUDENCE regions. Total number of grid cells with changes > 0.01 is given in parentheses.

|  | cropland | irri. cropland | grassland | forest | urban |
|---|---|---|---|---|---|
| Europe | 98.9% (96667) | 98.5% (26997) | 94.4% (86499) | 95.4% (32390) | 100% (13541) |
| Mid-Europe (ME) | 99.5% (7484) | 99.3% (861) | 85.4% (6967) | 97.6% (1647) | 100% (2178) |
| France (FR) | 99.5% (3706) | 100% (801) | 82.5% (3682) | 97.5% (1092) | 100% (443) |
| Mediterranean (MD) | 99.4% (5267) | 98.4% (3015) | 93.2% (5223) | 96.2% (1538) | 100% (1027) |
| British Isles (BI) | 99.9% (3710) | 100.0% (27) | 86.4% (4018) | 92.0% (3156) | 100% (950) |
| Iberian Peninsula (IP) | 98.7% (6535) | 98.0% (3073) | 94.0% (5829) | 96.6% (1532) | 100% (723) |
| Eastern Europe (EA) | 97.0% (13559) | 98.0% (1055) | 96.9% (13800) | 86.1% (2171) | 100% (2542) |
| Scandinavia (SC) | 99.5% (8643) | 98.1% (486) | 92.6% (4775) | 98.9% (5324) | 100% (807) |
| Alps (AL) | 95.6% (2560) | 91.6% (1029) | 95.6% (3407) | 98.1% (3106) | 100% (1061) |

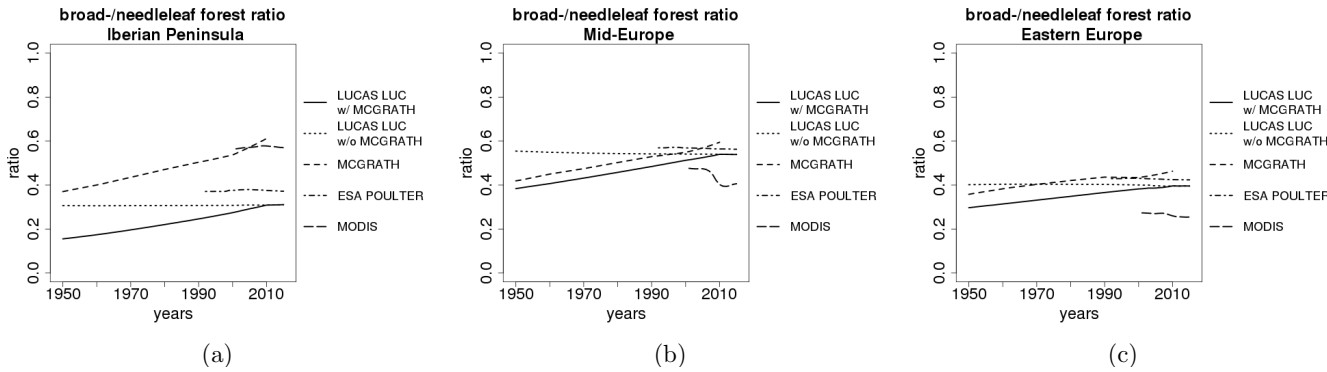

(a)  (b)  (c)

**Figure 5.** Ratio of broad- and needleleaf PFTs computed for LUCAS LUC with and without McGrath forest type data, ESA POULTER and MODIS for the PRUDENCE regions a) Iberian Peninsula, b) Mid-Europe, and c) Eastern Europe.

LUCAS LUC with McGrath employed are close to the trends in the original McGrath dataset while the absolute values differ. The two satellite-based datasets, ESA POULTER and MODIS, do not show strong trends in the broad-/needleleaf ratio until 2010 (last year of the McGrath dataset). For Eastern Europe, the trends are opposite with a slight increase in the broad-/needleleaf ratio in ESA POULTER and MODIS and a larger decrease in McGrath and LUCAS LUC.

## 3.2  Future LULC

Future changes in land cover fractions between 2015 and 2100 for the eight different scenarios (Table A2) are presented in Fig. 7-11. Aggregated area changes for the three PRUDENCE regions IP, ME, and EA as well as for Europe (25°W-50°E, 30°N-70°N) are shown in Fig. 12-15.



**Figure 6.** Changes in grid cell fraction and the spread of the changes (Sect. 2.4) for a,b) cropland, c,d) irrigated cropland, e,f) grassland, g,h) forest and i,j) urban classes based on LUCAS LUC between 1950 and 2015. Only the changes are show, where all LUCAS LUC reconstructions agree in the sign of the change (Sect. 2.4).





### 3.2.1 Cropland

While cropland fractions decrease in the historical period from 1950 to 2015 over most of Europe (Fig. 6a), a continent-wide decrease until 2100 is projected only for the SSP3/RCP7.0 scenario (Fig. 7h), with the exception of France, Turkey, and Belarus. The two SSP1-based scenarios show very similar patterns of cropland changes with strong expansion in Western Russia and large decreases in Southern Russia, Ukraine, Hungary, England, Denmark, and Northern Germany. The strongest increase in cropland is found for the SSP4/RCP3.4 scenario. The SSP5/RCP8.5 scenario shows small changes over Europe and

large cropland decreases in Northern Africa and the Middle East. Large block-like features are visible in Scandinavia, Russia and Turkey for most scenarios, which have an extent of about 2°, which are likely caused by the LUH2 workflow (Sect. 4.2).

The temporal evolution of aggregated cropland area shows that the changes are not steady for all scenarios (Fig. 12). For instance, the cropland area for Europe and in particular for the ME region shows rapid increase from 2050 onwards in the SSP5/RCP3.4OS while staying rather constant before (Fig. 12a,c). The two SSP1 scenarios diverge in their evolution around

2025 for Europe and the IP and ME regions but rather converge at the end of the century (Fig. 12a,b,c). While showing a slight decrease in cropland cover aggregated for Europe, the cropland area stays almost constant in the three PRUDENCE regions for SSP5/RCP8.5.

### 3.2.2 Irrigated cropland

For the future scenarios, the irrigated cropland fractions show different signals depending on the region (Fig. 8).Signals for

SSP1/RCP1.9, SSP1/RCP2.6, and SSP3/RCP7.0 (Fig. 8a,b,g) go into the same direction (increase) but with different magnitudes. Whereas the SSP1-based scenarios show small changes, the SSP3/RCP7.0 scenario projects large changes predominantly in the Middle East (Urmia Basin) and Turkey where a strong increase in irrigated cropland is expected.

The strongest changes for Europe are projected by the SSP5/RCP3.4OS scenario which shows a continent-wide increase in irrigated cropland (Fig. 8d) and of cropland in general (Fig. 7d). Exceptions, such as regions along the Po river in Italy and

along the Euphrates and Tigris in Iraq show a decrease of irrigated cropland for this scenario.

In contrast to the continent-wide increase in irrigated cropland in the scenario SSP5/RCP3.4OS, the SSP4-based scenarios with the pathways RCP3.4 (Fig. 8c) and RCP6.0 (Fig. 8f) project a continent-wide decrease of irrigated cropland, with some exceptions around the Mediterranean Sea, along Africa's west coast and in Central Asia.

Most scenarios agree on the change signal in multiple regions. Whereas regions along the Euphrates and Tigris in Iraq, along

the Nile in Egypt and along the Po River in Italy expect a decrease of irrigated cropland in most scenarios, regions in the North Caucasus and in the Urmia Basin in Iran show an increase in irrigated cropland in most scenarios.

### 3.2.3 Grassland

The SSP1-based scenarios as well as SSP4/RCP3.4 and SSP5/RCP3.4OS (Fig. 9a-d) show a strong decrease in grassland cover for most of Europe. For SSP1/RCP1.9 and SSP1/RCP2.6 this is due to the conversion of grassland to forest 10a,b) while for

SSP4/RCP3.4 and SSP5/RCP3.4OS grassland is mainly converted to cropland (Fig. 9c,d). Similarly, the increase in grassland



Earth System
Science
Data

in Southern Russia in the SSP4/RCP3.4 scenario is due to the conversion from cropland to grassland. The grassland decrease is not as strong in the SSP2/RCP4.5 and SSP4/RCP6.0 scenarios where also large regions with increase are visible (Fig. 9e,f). The SSP3/RCP7.0 scenario shows a number of regions with a large grassland cover increase such as Spain, Germany, Norway, the Alps, and Carpathian Mountains (Fig. 9g). The increase in Norway and in the Alps is compensated by deforestation (Fig. 10g). Almost no changes in grassland cover over Europe are visible for the SSP5/RCP8.5 scenario (Fig. 9h). However, a large increase is found in Northern Africa and the Middle East, which is compensated by a decrease in cropland (Fig. 7h). The block-like structures that are found for the cropland changes are also visible for the grassland changes.

Except for SSP5/RCP3.4OS, which shows a strong abrupt decrease from 2050 to 2055, increase and decrease in grassland cover over Europe is steady from 2015 onwards (Fig. 13a). The abrupt change in SSP5/RCP3.4OS is even more pronounced in the ME and EA regions (Fig. 13c,d). In the latter region, also the SSP4/RCP3.4 scenario shows a steep decline in grassland cover after 2050. While grassland cover strongly increases in one scenario (Fig. 13b) in the IP region grassland cover either stays almost constant over time or decreases in the other two regions.

### 3.2.4 Forest

A strong forest cover increase is found for SSP1/RCP1.9, SSP1/RCP2.6 (Fig. 10a,b) and to a lesser extent for SSP2/RCP4.5 (Fig. 10e). These are the scenarios for which the added tree cover fraction files are used because the original LUH2 dataset underestimates the strong afforestation signal (Sect. 2.3.2). In the SSP1-based scenarios, the largest increase is found in Ireland, England, Northern France, and in Russia near the border to Ukraine. Also, a widespread increase is visible for most countries in Central and Eastern Europe, Southern Sweden and Finland, and Northern Spain. The forest increase is mainly compensated by a decrease in grassland (Fig. 9) and shrubland (not shown) and to a lesser extent by declining cropland (Fig. 7). The only region with substantial forest reduction is Western Russia. This decrease is also visible in the SSP4/RCP3.4, SSP5/RCP3.4OS, and SSP2/RCP4.5 scenarios (Fig. 10c-e). As for cropland and grassland, SSP5/RCP8.5 shows only small forest cover changes (Fig. 10h)

The aggregated forest cover changes for Europe show the steep increase from 2016 onwards (Fig. 14a). In the ME region, the forest cover increase levels off around 2050 while the IP and EA regions show steady increases (Fig. 14b-d). Especially in the ME and EA regions, the magnitude of the increase is many times larger than for the changes in the historical period. In contrast, the afforestation in the SSP2/RCP4.5 scenario starts in 2050 and continues until 2100 with a magnitude comparable to historical changes. A substantial deforestation signal in the SSP2/RCP4.5 is visible in the IP and EA region from 2050 onwards.

### 3.2.5 Urban

In contrast to the historical period, all scenarios show both decreases and increases in urban fraction between 2015 and 2100 (Fig. 10). Urban changes are largely driven by the SSP scenarios (i.e. population dynamics) resulting in almost identical changes in the LUCAS LUC dataset in scenarios based on the same SSP scenario. A widespread urbanization signal can be found in the SSP5-based scenarios for Europe except for Eastern Europe, which shows a decrease in urban fraction (Fig.



10d,h). The increase in urban fractions is particularly strong in Great Britain. The SSP1-based scenarios show a increase with
a smaller magnitude in West- and South Europe, respectively, as well as Scandinavia and a decrease in the eastern European
countries including some parts of Germany (Fig. 11a,b). Based on the SSP4/RCP6.0 scenario, only a few urban areas in Spain,
France, Italy, England, and the Czech Republic exhibit an increase in urban fraction. In East- and Central Europe urban fraction
decreases, with Germany experiencing the largest decrease. The SSP3/RCP7.0 scenario is the only scenario which does not
show a decrease in urban fraction in Russia. Instead, it shows decreases in Western and Central European countries.

The time series analysis of the aggregated changes for Europe shows that all scenarios project an increase in urbanization
until 2050 (Fig. 15a). Only in the SSP5-based scenarios the total urban area increases further until 2100. For the other scenarios,
urban fraction remains constant or declines. However, there are regional differences. In the EA region, all scenarios show a
peak in urbanization until 2050 and a decreases until 2100 with the exception of SSP3/RCP7.0, where the peak is already
reached by 2040 (Fig. 15d). The IP and ME regions show a similar temporal evolution of urban areas (Fig. 15b,c). However,
the decrease is stronger in ME. Here, the total urban area in 2100 is even smaller than the total area in 1950 in the SSP3/RCP7.0
scenario.

# 4 Discussion

## 4.1 Uncertainties

The uncertainties of the LUCAS LUC dataset with respect to the historical changes were subdivided into uncertainty in the
underlying base map for 2015 (i.e., LANDMATE PFT map), the uncertainties in the LUH2 datasets and the different resolution
between the LANDMATE PFT map (0.1°) and the LUH2 dataset (0.25°).

    The uncertainty of the present-day pattern of the land cover (i.e., the LANDMATE PFT map) was assessed for Europe in
the companion paper (Reinhart et al., 2022b). The ESA-CCI LC dataset, which is the baseline for the LANDMATE PFTs, has
been previously validated globally (e.g., Hua et al., 2018) and in a regional approach limited to Eastern Europe (e.g., Reinhart
et al., 2021). Depending on the validation method, which is a limiting factor of such a validation assessment, the ESA-CCI
LC dataset was shown to be of a very good quality for the dominant land cover classes cropland and forest but certain issues
were found for other classes. Reinhart et al. (2021) showed an overall accuracy of 76% for the ESA-CCI LC dataset in Eastern
Europe, where cropland and forest showed >81% accuracy, respectively, but accuracy values lower than 50% for the other
categories assessed. Some shortcomings of ESA-CCI LC could be overcome through targeted varying of the LANDMATE
PFT Cross Walking Procedure (CWP). For example, the known too small shrub proportions of ESA-CCI LC over Europe were
partly compensated by increasing the shrub proportions for certain land cover class translations into PFTs.

    Therefore, compared to ESA POULTER, the map generated using the CWT by Poulter et al. (2015), the LANDMATE PFTs
were improved for shrubland, forest, grassland, and bare area cover, respectively, and slightly worse for cropland, which is
discussed in the associated publication by Reinhart et al. (2022b). The overall accuracy of the LANDMATE PFT dataset is
about 73%.





**Figure 7.** Changes in grid cell cropland fraction based on LUCAS LUC for the RCP/SSP scenarios a) SSP1/RCP1.9 b) SSP1/RCP2.6, c) SSP4/RCP3.4, d) SSP5/RCP3.4OS, e) SSP2/RCP4.5, f) SSP4/RCP6.0, g) SSP3/RCP7.0, h) SSP5/RCP8.5 between 2015 and 2100.

The uncertainties from LUH2 dataset are discussed by Hurtt et al. (2020) and the additional uncertainty information, in form of two different historical reconstructions, were employed in the present study to quantify the impact of this uncertainty on the historical changes within LUCAS LUC. The results of this analysis show that this uncertainty increases with time and that the different LUH2 reconstructions have the largest contribution to the spread of the changes except for urban land cover, where

the resolution differences is more important. Overall the majority of the changes between 1950 and 2015 are larger than the

**Figure 8.** Same as Fig. 7 but for irrigated cropland PFTs.

spread with fraction of robust changes of 90% and higher for most land cover classes and regions in Europe. Only grassland and forest changes are lower for some regions with lowest values for grassland changes in FR (82.5%).

The differences in historical LULCC between the LUCAS LUC and other LULCC datasets are mainly caused by the differences between LUH2 and the other datasets due to the close connection between LUCAS LUC and LUH2 LULCC. Li et al. (2018), who compared ESA-CCI LC with LUH2 and other available global datasets, attributed these differences (1) to the treatment of shifting cultivation, (2) to the still coarse resolution of ESA-CCI LC, which limits the detection of land cover



**Figure 9.** Same as Fig. 7 but for grassland PFTs.

changes to larger scale changes, and (3) to the difference between the inventory-based approach (i.e. countries report to the land use statistics to the FAO) taken by LUH2 and the satellite-based approach taken by ESA-CCI LC. For instance, Keenan et al. (2015) noted that cleared forest due to wood harvesting is not reported as forest loss if secondary forest is planted because the land-use does not change.


Differences between LUCAS LUC and the two datasets ESA POULTER and HILDA+ are most pronounced with respect to forest changes. Kuemmerle et al. (2016) noted that satellited-based datasets include naturally driven changes e.g. due to

**Figure 10.** Same as Fig. 7 but for tree PFTs.

forest fire and wind storms as well as management driven changes e.g. due to wood harvesting. Ceccherini et al. (2020) showed that averaged over Europe such changes are small compared to forest harvesting but can be larger in regions with frequent

forest fires (e.g., Portugal). Also the different definition of forest in LUH2, which is based on a biomass density threshold, and ESA-CCI LC, based which is based on tree cover, could have a substantial impact on the forest transitions. For instance, the afforestation signal over the Iberian Peninsula is mainly due to farmland abandoning and the regrowth of natural vegetation (Vilà-Cabrera et al., 2017; Palmero-Iniesta et al., 2021). A detailed analysis of the ESA POULTER time series is needed to

**Figure 11.** Same as Fig. 7 but for urban.

investigate if these processes caused the discrepancies in forest changes LUH2/LUCAS LUC and HILDA+/ESA-CCI LC.

Another possibility is the already mentioned uncertainty originating from the CWP. However, the impact of the CWP on the computed land cover changes has not been analyzed so far.

Notable differences between LUCAS LUC and ESA POULTER/HILDA+ are also found in the urban land cover changes. Here, ESA-CCI LC seems to largely overestimate the rate of urbanization in Europe between the late 1990s and the early 2000s, whereas urban land cover changes in LUCAS LUC seem to be more reasonable during this period. The other satellite-

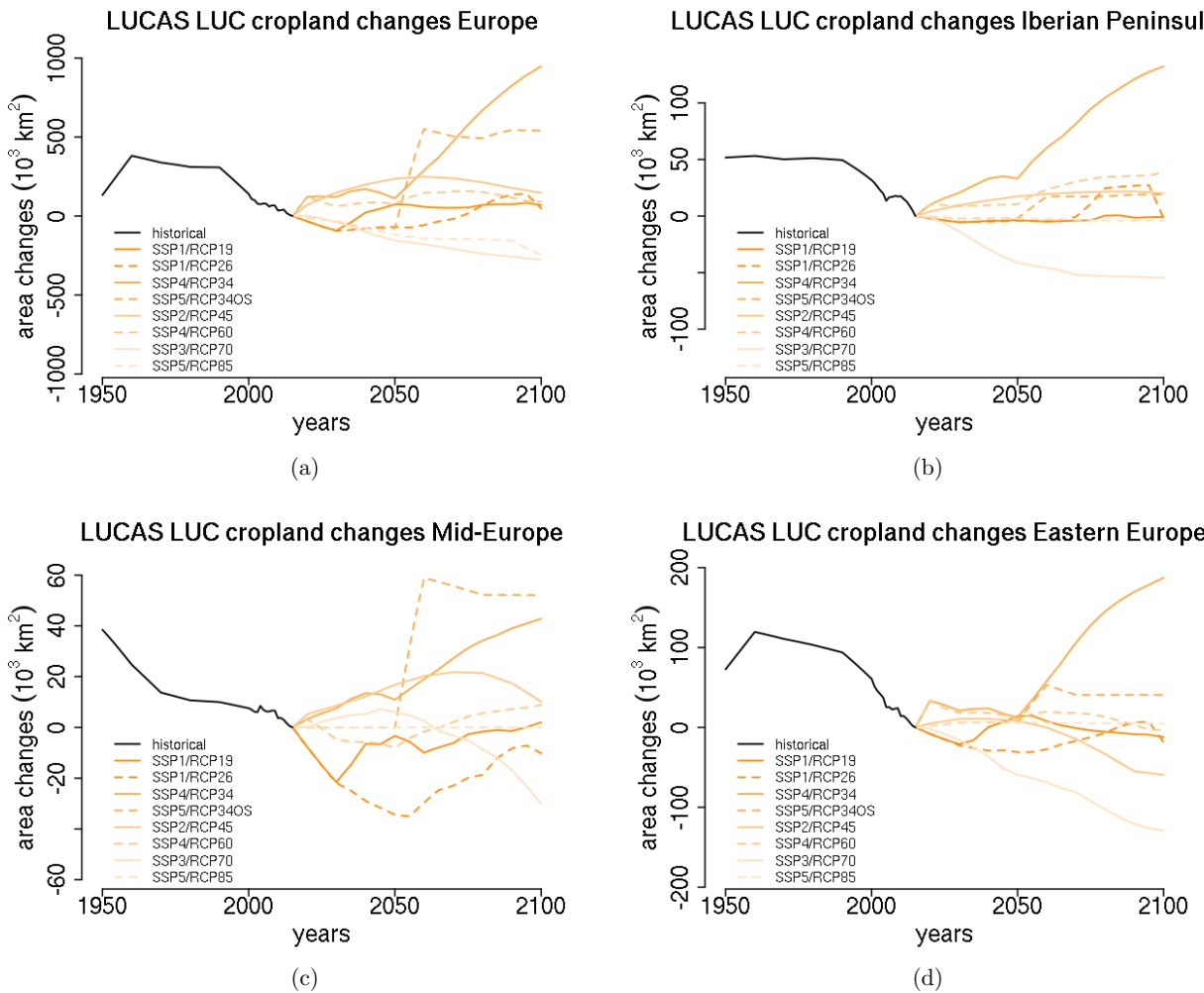

**Figure 12.** Area changes with respect to the year 2015 in cropland PFTs computed for LUCAS LUC for a) Europe and the PRUDENCE regions b) Iberian Peninsula, c) Mid-Europe, and d) Eastern Europe.

based PFT time series generated from MODIS shows much larger annual land cover changes, which for some regions seem questionable. On the other hand, urban cover changes in MODIS are likely too small. Güneralp et al. (2020) showed that based on a literature review the average increase in urban land cover ranges between 2 to 3% per year in Europe in the 1990s and 2000s. The large urban changes in HILDA+ could be caused by the difference in the definition of urban and therefore lead to a different extent of the urban areas. For instance, the HILDA+ urban land cover in ME in 2015 (approx. $90x10^3 \ km^2$)

is more than twice as large than MODIS, ESA POULTER, and LUCAS LUC (approx. $40x10^3 \ km^2$). Reinhart et al. (2021) and Demuzere et al. (2019) showed that ESA-CCI LC underestimates urban land cover compared to CORINE and LCZ maps, respectively, because it misses low-rise built up areas.






**Figure 13.** Same as Fig. 12 but for grassland PFTs.

Given the uncertainties and issues with respect to the other LULCC datasets and based on the detailed analysis of the historical land cover changes in the results section, LUCAS LUC land cover changes are reasonable for the historical period.

### 4.2 Indented use and limitations

The newly generated LULCC dataset LUCAS LUC is tailored towards the requirements of future CMIP6 downscaling experiments within the FPS LUCAS and EURO-CORDEX. The need for high-resolution land cover input is met by employing the ESA-CCI LC dataset, which has a ~300 m grid, as a basemap for the year 2015. Since most of the state-of-the-art LSM employ





**Figure 14.** Same as Fig. 12 but for tree PFTs.

a PFT land cover classification the ESA-CCI LC was converted into PFTs. This step also helps dealing with mixed ESA-CCI
LC land cover classes, which can be conveniently converted into classes with similar properties.

As indented, LUCAS LUC land cover changes closely follow the transitions for cropland, urban, and forest provided by
LUH2 with some exceptions discussed in the previous section. Hence, by employing the LUCAS LUC dataset the land use
and land cover forcing of the RCMs is consistent with the forcing of the driving CMIP6 GCM data. However, some GCMs
do not use all land use transitions leaving the transitions of natural vegetation to their dynamic vegetation models. LUCAS
LUC changes are generally slightly smaller for all three land cover types in comparison to the LUH2 changes but follow the
temporal evolution of LUH2. This can be attributed to the LUT, which keeps the bare ground fraction constant in LUCAS LUC,
which limits the possible land cover changes. This was done because LUH2 does not provide information about changes in

**Figure 15.** Same as Fig. 12 but for urban.

bare areas. Thus, desertification, urban expansion or cropland expansion into desert areas are not included in LUCAS LUC. For Europe, those land cover conversions are not common. However, for regions with large desert areas (e.g., Northern Africa and the Mid-East) this limitation could substantially underestimate land cover changes. In addition, the difference between LUH2 and LUCAS LUC can be partially attributed to the computation of the land cover area from the PFT fractions. For LUCAS LUC, the ESA-CCI LC land-sea mask was used, which also includes rivers and lakes, while this is not the case for the LUH2 land-sea mask. Consequently, it is likely that smaller total area changes are computed in LUCAS LUC compared to LUH2 in regions with lakes and rivers as well as near coastlines.

The main LUCAS LUC land cover change signals for Europe between 1950 and 2015 are the reduction of cropland but with an extension of irrigated cropland, afforestation in mountainous areas, and urbanization. The magnitude and the spatial





extent of these changes are considerable and are therefore likely to affect the simulated European climate. Even the smaller land cover changes within the ESA-CCI LC dataset altered the climate change signal simulated with RCM for the period 1992 to 2015 (Huang et al., 2020). However, as discussed before, LUCAS LUC deviate from other available LULCC datasets for some regions and some land cover classes, while the LULCC datasets considerably deviate from each other, too. This needs to be considered, when analysing and evaluating the downscaled model results for particular regions for the historical period. For assessing the sensitivity of the RCM results to the LULCC input, additional ensemble experiments could be set-up employing different historical LULCC datasets.

In addition to other LULCC datasets, LUCAS LUC also provides historical changes in the broad-/needleleaf forest ratio. The conversion of broadleaf forests to needleleaf forests in Europe are not observed by ESA POULTER or MODIS, which might be due to the differences in satellite-based and inventory-based datasets. This discrepancy should be investigated in the future. The irrigated cropland increase in LUCAS LUC is also substantial and is likely to be relevant when investigating historical changes in the climate of Southern Europe, where the largest increase occurs. However, the LUCAS LUC dataset does not distinguish between irrigation methods (e.g., sprinkler irrigation, channel irrigation) which might show different effects on the climate (Valmassoi et al., 2019). Hence, there is a need for high-resolution European-wide dataset with information on the distribution and the development of irrigation methods.

Future land cover changes are even larger than the historical changes for some of the available scenarios. Hence, substantial policy changes would be necessary to reach the amount of land cover conversions in the densely populated Europe, where land ownership is both public and private. In addition, there are large regional differences. The two SSP1-based scenarios, which are the low-end scenarios, show a strong afforestation signal compensated by a decrease in grass- and shrubland. Also, noticeable changes in cropland (both increase and decrease) are projected for these scenarios. Hence, the LULCC induced climate change signal might be comparable to the greenhouse-gas induced signal in regions with large LULCC for some seasons (Hirsch et al., 2018). For instance, Davin et al. (2020) showed that for a extreme afforestation scenario temperature changes simulated with RCMs can range up to +/-2 K in Europe in the summer season. This emphasize the need to include LULCC when downscaling GCM/ESM projections based on these scenarios. Interestingly, the high-end scenario SSP5/RCP8.5 shows the smallest land cover changes except for urbanization, where it has the largest signal together with the other SSP5-based scenario (i.e. SSP5/RCP3.4OS). Therefore, it might be harder to detect LULCC induced regional climate changes giving the strong greenhouse gas forcing. In contrast, the SSP3/RCP7.0 scenario, which has also a large greenhouse gas forcing, shows large-scale cropland decreases and regions with deforestation (e.g., Alps and Scandinavia) as well as afforestation (e.g., Po Valley and Carpathian Mountains).

It needs to be noted, that the future land cover changes provided by LUCAS LUC consider anthropogenic land use changes, but do not account for potential latitudinal and altitudinal shifts of the of natural vegetation or in particular forest due to climate change (McDowell et al., 2020) because the underlying LUH2 data only provides land use changes due to anthropogenic activities. Therefore, the potential northwards expansion of forest in Europe, which is projected under different climate changes scenarios (Dyderski et al., 2018), is not included in LUCAS LUC. Furthermore, in contrast to the historical LUCAS LUC reconstruction, the future forest composition does not change because the relative fractions of the tree and shrub PFTs stay

constant during the forward translation. However, both the shift in the composition and the spatial distribution depend on the projected climate by the different ESMs/GCMs and are therefore uncertain.

The large block-like features appearing in the cropland and grassland change signals in all scenarios might be attributed to the harmonization process within the LUH2 workflow. Annual changes in cropland, grazing land, and urban areas are computed and aggregated to a 2° grid and subsequently disaggregated to the final 0.25° grid (Hurtt et al., 2020). It is therefore likely that the disaggregation step did not fully dissolve the grid structure of the coarse 2° grid. For GCMs or ESMs with a typical resolution of around 1° this might not have caused any issues. However, for RCMs with a typical resolution of about 0.1°, for which the LUCAS LUC dataset has been created, the impact of such structures in the LULCC needs to be carefully
investigated.

## 5    Data availability

The LUCAS LUC historical land use and land cover change dataset (Version 1.1) and the LUCAS LUC future land use and land cover change dataset (Version 1.1) are published with the Long Term Archiving Service (LTA) for large research datasets which are relevant for climate or Earth system research of the German Climate Computing Service (DKRZ). The DKRZ LTA
is accredited as regular member of the World Data System. Both datasets are available within the LANDMATE project data at https://www.wdc-climate.de/ui/entry?acronym=LUC_hist_EU_v1.1 (Hoffmann et al., 2022b) and https://www.wdc-climate. de/ui/entry?acronym=LUC_future_EU_v1.1 (Hoffmann et al., 2022a). Within the LANDMATE project, a short documentation summarizes the technical information on the LANDMATE PFT and LUCAS LUC dataset.

## 6    Conclusions

The need of the RCM community for a high-resolution LULCC dataset is met using high-resolution PFT maps based on the ESA-CCI LC dataset and land use change information from the LUH2 dataset that were translated into PFT changes using a newly developed LUT. The resulting LUCAS LUC dataset is tailored towards RCM requirements. Urbanization, which is mostly discarded by LUTs, is included as well as changes in irrigated cropland. For the historical period, also changes in the broad-/needleleaf forest ratio are considered employing an additional forest type dataset by McGrath et al. (2015).
The LUCAS LUC dataset enables RCM modellers to include historical and future annual LULCC into the next-generation downscaling experiments (e.g., within FPS LUCAS and EURO-CORDEX) based on CMIP6 projections. Consequently, the impact of LULCC on the regional climate change signals can be investigated. For most of Europe, past and future trends in cropland, forest, and urban areas in LUCAS LUC are consistent with the LUH2 dataset albeit with a slight underestimation of the magnitude. A comparison with other global datasets revealed substantial differences in the trend of some land cover
classes. However, the differences between the ESA-CCI LC based dataset and the MODIS-based dataset are also quite large showing the uncertainty related to the approaches employed to estimate LULCC.



Earth System
Science
Data

The future LULCC for the eight SSP/RCP scenarios show substantial changes that can exceed the observed historical LULCC in Europe. Hence, the regional climate change signals, simulated by RCMs, are likely to be affected by these changes and should, therefore, be considered in upcoming downscaling experiments. Especially when downscaling projections for the low-end scenarios (i.e. SSP1/RCP1.9 and SSP1/RCP2.6), which show a strong afforestation signal, the biogeophysical effect of LULCC is expected to be of the order of the greenhouse-gas induced effects in some regions (Hirsch et al., 2018). In contrast, for the high-end scenario SSP5/RCP8.5 LULCC in Europe are small compared to the other scenarios except for urbanization.

While the current dataset is provided on a 0.1° grid for Europe in order to be suited for the EURO-CORDEX EUR-11 grid, the method could be applied to generate data at even higher resolution, e.g., needed for convective permitting RCM experiments (Coppola et al., 2020). However, a downscaling of the coarse land-use changes provided by LUH2 would be necessary, e.g. by using a spatial disaggregation model (Chen et al., 2020).

The LUCAS LUC dataset can also be prepared for other CORDEX regions because most of the input data is provided globally but with limitations to certain land cover changes such as desertification, which cannot be considered because bare area changes are not available from LUH2. The LUCAS LUC datasets were already produced for other CORDEX regions (Hoffmann et al., 2021) are currently validated. The quality of conversion of ESA-CCI LC classes into PFTs might to some extent depend on the availability of high-resolution climate data, needed for the Holdridge-based cross-walking procedure and data on forest type conversion for the historical period might not be available for other regions.

*Author contributions.* PH developed the workflow with DR, VR, NdN, ELD, JB, and BB. PH developed the land use translator with DR, NdN, and ELD. VR and PH developed the cross-walking procedure. PH wrote the code for the land use translator and for the processing of different datasets and generate the LUCAS LUC dataset. SL provided the McGrath dataset. VR generated the ESA POULTER with the help of the ESA-CCI user tool, downloaded and prepared the MODIS PFT dataset. CA wrote the sections on analysis of irrigated cropland (i.e. Sect. 3.2.2 and 3.1.2) and prepared table A2. VR wrote the description of the ESA POULTER and MODIS datasets (i.e. Sect. 2.5.1 and Sect. 2.5.2), visualized the LANDMATE PFTs and prepared tables A3 and A4. PH conducted the analysis of the land cover changes, visualized the results, and wrote all other sections. All authors reviewed the paper draft and contributed to the final paper.

*Competing interests.* The authors declare that they have no conflict of interest.

*Acknowledgements.* This work was financed within the framework of the Helmholtz Institute for Climate Service Science (HICSS), a co-operation between Climate Service Center Germany (GERICS) and Universität Hamburg, Germany and conducted as part of the project LANDMATE (Modelling human LAND surface modifications and its feedbacks on local and regional cliMATE). We acknowledge the sup-



port of LUCAS by WCRP-CORDEX as a Flagship Pilot Study. We acknowledge the E-OBS dataset from the EU-FP6 project UERRA (http://www.uerra.eu) and the Copernicus Climate Change Service, and the data providers in the ECA&D project (https://www.ecad.eu). We thank the European Space Agency (ESA) for making the Land cover products publicly available. Special thanks go to the FPS LUCAS partners for providing useful comments in order to improve the dataset.





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



**Table A1.** List of Abbreviations.

| Abbreviation | Meaning |
| --- | --- |
| AL | PRUDENCE region Alps |
| BI | PRUDENCE region British Isles |
| CMIP6 | Coupled Model Intercomparison Project phase 6 |
| CORDEX | Coordinated Downscaling Experiment |
| CORINE | Coordination of Information on the Environment |
| CRU | Climatic Research Unit |
| CWP | cross walking procedure |
| CWT | cross walking table |
| E-OBS | European daily high-resolution gridded dataset |
| EA | PRUDENCE region Eastern Europe |
| ESA POULTER | plant functional type dataset based on ESA-CCI LC using the Poulter et al. (2015) cross walking tables |
| ESA-CCI LC | European Space Agency Climate Change Initiative Land Cover |
| ESM | earth system model |
| EUR-11 | EURO-CORDEX domain at 0.11° resolution |
| EUR-44 | EURO-CORDEX domain at 0.44° resolution |
| EURO-CORDEX | Coordinated Downscaling Experiment - European Domain |
| FAO | Food and Agriculture Organization of the United Nations |
| FPS | flagship pilot study |
| FR | PRUDENCE region France |
| GCM | global climate model |
| GLM2 | Global Land Use Model |
| GlobCover | Global Land Cover Map |
| HICSS | Helmholtz Institute for Climate Service Science |
| HILDA+ | global HIstoric Land Dynamics Assessment + |
| HLZ | Holdrige Life Zones |
| HYDE 3.2 | History Database of the Global Environment version 3.2 |
| IAM | integrated assessment model |
| IP | region Iberian Peninsula |
| LANDMATE | HICSS project "Modelling human LAND surface Modifications and its feedbacks on local and regional cliMATE" |
| LANDMATE PFT | LANDMATE plant functional type dataset |
| LSM | land surface model |
| LUCAS | WCRP CORDEX Flagship Pilot Study Land Use and Climate Across Scales |
| LUCAS LUC | LUCAS Land Use and land Cover change dataset |
| LUH2 | Land-Use Harmonization 2 |
| LULCC | land use and land cover change |
| LUMIP | Land Use Model Intercomparison Project |
| LUT | land use translator |
| MD | PRUDENCE region Mediterranean |
| ME | PRUDENCE region Mid-Europe |
| MODIS | Moderate Resolution Imaging Spectroradiometer |
| MsTMIP | Multi-scale Synthesis and Terrestrial Model Intercomparison Project |
| NACP | North American Carbon Program |
| NCAR | National Center for Atmospheric Research |
| PFT | plant functional type |
| PRUDENCE | Prediction of Regional scenarios and Uncertainties for Defining EuropeaN Climate change risks and Effects |
| RCM | regional climate model |
| RCP | Representative Concentration Pathways |
| SC | PRUDENCE region Scandinavia |
| SSP | Shared Socioeconomic Pathways |
| WCRP | World Climate Research Program |



**Table A2.** Specfication of the land-use change scenarios provided by LUH2 (Hurtt et al., 2020) and the assumptions for land-use and land cover developments in the different scenarios (Hurtt et al., 2020; Popp et al., 2017; Riahi et al., 2017; van Vuuren et al., 2011)

| SSP | RCP | IAM | Short summary of scenarios |
|---|---|---|---|
| 1 | 1.9 | IMAGE | Green growth paradigm<br>• reaches a maximal global warming of 1.5°C<br>• low - moderate population growth<br>• high economic growth<br>• respected environmental boundaries & regulated land use which avoids deforestation & supports restoration of forests<br>• healthy diets with low animal-calories shares |
| 1 | 2.6 | IMAGE | Green growth paradigm<br>• follows SSP1 but reaches a maximum global warming of 2°C |
| 4 | 3.4 | GCAM | Intermediate pathway<br>• high inequalities between societies<br>• use of bioenergy leads to a large-scale increase in cropland<br>• regulated land use & afforestation in high - medium-income countries<br>• deforestation due to cropland expansion in low-income countries |
| 5 | 3.4OS | REMIND-MAGPIE | Overshoot scenario<br>• target level of global is overshoot, followed by a strong mitigation strategy<br>• no mitigation and fossil fuel based developments till 2040<br>• strong mitigation actions from 2040, which result in net negative CO2 emissions in 2100<br>• use of bioenergy leads to a large-scale increase in cropland |
| 2 | 4.5 | MESSAGE-GLOBIOM | Intermediate pathway<br>• little shift from historical patterns<br>• moderate population growth<br>• partly regulated land use<br>• inequality in societies<br>• international cooperation for mitigation are delayed till 2040 |
| 4 | 6.0 | GCAM | Inequalities<br>• environmental policies & regulated land use leading to increase in cropland, pasture & forest in high/medium-income countries<br>• low agricultural productivity in low-income countries |
| 3 | 7.0 | AIM | Regional rivalry<br>• focus is on regional development<br>• high inequalities<br>• limited transfer of agricultural technologies leads to low agricultural intensification in developing countries<br>• unhealthy diets with high animal-calorie shares<br>• population growth is low in industrialized countries<br>• expansion of cropland & pasture into forest leads to a large-scale deforestation |
| 5 | 8.5 | REMIND-MAGPIE | Fossil fuel development<br>• high but resources intensive development<br>• doubled food demand<br>• expansion of cropland into pasture and forest<br>• no mitigation |



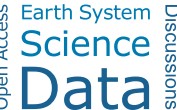

**Table A3.** MODIS plant functional types based on Bonan et al. (2002).

| PFTs | Names |
| --- | --- |
| 0 | Water bodies |
| 1 | Evergreen Needleleaf Trees |
| 2 | Evergreen Broadleaf Trees |
| 3 | Deciduous Needleleaf Trees |
| 4 | Deciduous Broadleaf Trees |
| 5 | Shrub |
| 6 | Grass |
| 7 | Cereal Croplands |
| 8 | Broadleaf Croplands |
| 9 | Urban and Built-up Lands |
| 10 | Permanent Snow and Ice |
| 11 | Barren |





**Table A4.** ESA CCI LC default cross-walking table for ESA CCI LC class translation to ESA PFTs.

| ESA LC class | 1 Tree broadleaf evergreen | 2 broadleaf deciduous | 3 needleleaf evergreen | 4 needleleaf deciduous | 5 Shrub broadleaf evergreen | 6 broadleaf deciduous | 7 needleleaf evergreen | 8 needleleaf deciduous | 9 Grass natural grass | 10 crop | 11 Non-vegetated bare ground | 12 water | 13 snow/ice | 14 urban[1] |
|---|---|---|---|---|---|---|---|---|---|---|---|---|---|---|
| 10 | | | | | | | | | | 100 | | | | |
| 11 | | | | | | | | | | 100 | | | | |
| 12 | | | | | | 50 | | | | 50 | | | | |
| 20 | | | | | | | | | | 100 | | | | |
| 30 | 5 | 5 | | | 5 | 5 | 5 | | 15 | 60 | | | | |
| 40 | 5 | 5 | | | 7.5 | 10 | 7.5 | | 25 | 40 | | | | |
| 50 | 90 | | | | 5 | 5 | | | | | | | | |
| 60 | | 70 | | | | 15 | | | 15 | | | | | |
| 61 | | 70 | | | | 15 | | | 15 | | | | | |
| 62 | | 30 | | | | 25 | | | 35 | | 10 | | | |
| 70 | | | 70 | | 5 | 5 | 5 | | 15 | | | | | |
| 71 | | | 70 | | 5 | 5 | 5 | | 15 | | | | | |
| 72 | | | 30 | | | 5 | 5 | | 30 | | 30 | | | |
| 80 | | | | 70 | 5 | 5 | 5 | | 15 | | | | | |
| 81 | | | | 70 | 5 | 5 | 5 | | 15 | | | | | |
| 82 | | | | 30 | | 5 | 5 | | 30 | | 30 | | | |
| 90 | | 30 | 20 | 10 | 5 | 5 | 5 | | 15 | | 10 | | | |
| 100 | 10 | 20 | 5 | 5 | 5 | 10 | 5 | | 40 | | | | | |
| 110 | 5 | 10 | 5 | | 5 | 10 | 5 | | 60 | | | | | |
| 120 | | | | | 20 | 20 | 20 | | 20 | | 20 | | | |
| 121 | | | | | 30 | | 30 | | 20 | | 20 | | | |
| 122 | | | | | | 60 | | | 20 | | 20 | | | |
| 130 | | | | | | | | | 60 | | 40 | | | |
| 140 | | | | | | | | | 60 | | 40 | | | |
| 150 | 1 | 3 | 1 | | 1 | 3 | 1 | | 5 | 85 | | | | |
| 151 | | 2 | 6 | 2 | | | | | 5 | 85 | | | | |
| 152 | | | | | 2 | 6 | 2 | | 5 | 85 | | | | |
| 153 | | | | | | | | | 15 | 85 | | | | |
| 160 | 30 | 30 | | | | | | | 20 | | | 20 | | |
| 170 | 60 | | | | 20 | | | | | | | 20 | | |
| 180 | | 5 | 10 | | | 10 | 5 | | 40 | | | 30 | | |
| 190 | | | | | | | | | | | | | | 100 |
| 200 | | | | | | | | | | | 100 | | | |
| 201 | | | | | | | | | | | 100 | | | |
| 202 | | | | | | | | | | | 100 | | | |
| 210 | | | | | | | | | | | | 100 | | |
| 220 | | | | | | | | | | | | | 100 | |