# Peer review of "High-resolution land use and land cover dataset for regional climate modelling: Historical and future changes in Europe"

_Earth System Science Data, 2022_

## Author Comment (AC1)

**Author Response to the Reviewer Comments to the manuscript "High-resolution land use and land cover dataset for regional climate modelling: Historical and future changes in Europe" [essd-2022-431] submitted to Earth System and Science Data.**

We thank David Carlson for coordinating the review process as well as Jason Evans and one anonymous referee for their very valuable reviews. As you will see from our detailed point-by-point responses below (show in black), we have carefully gone through all of the reviewers' comments and suggestions (shown in red). The changes discussed in this reply will be included in the revised manuscript and dataset and will thus become visible after re-submission.

**Response to Reviewer 1**

*This paper describes a land cover change dataset for use with regional climate models run for any period from 1950 to 2100 over the Euro-CORDEX region. Future land cover maps are derived for each of the SSP scenarios and are based on the Land-Use Harmonization 2 (LUH2) dataset. This dataset is an important enabler of downscaled climate projections using the SSP scenarios such as those under the CORDEX-CMIP6 project.*

*The article is appropriate to support the dataset. It presents the methodology in sufficient detail to enable replication and evaluates the dataset against multiple products and includes an estimate of the uncertainty.*

*The dataset was easily accessible and downloadable after a registration step. It comes as netCDF format files making it very easy to use with the appropriate metadata embedded in the files.*

*Overall the dataset is unique, useful and complete for the Euro-CORDEX region. It would be great to see this same methodology be applied to produce similar datasets for all CORDEX regions globally at some point in the future.*

*The paper is well written and worthy of publication. I have only technical corrections that should be addressed before publication below.*

Thank you very much for this positive feedback and pointing out our typos. We are indeed exploring the extension of the LUCAS-LUC dataset to other regions around the globe.

All suggested technical corrections were implemented:

*Technical corrections*

1. *ln 72: delete extra "keeping"*
2. *ln 399: "lager" should be "larger"*
3. *ln 546: delete the first "based"*
4. *ln 609:"emphasize" should be "emphasizes"*
5. *ln 621: "giving" should be "given"*
6. *ln 619: "changes" should be "change"*
7. *ln637: "documentation" should be "document"*
8. *ln 556:"currently validated" should be "currently being validated"*

**Response to Reviewer 2**

*This preprint describes a newly developed high-resolution land use/cover dataset for Europe, ready-to-use for Regional Climate Models. The presented dataset represents a novelty, since it provides higher spatial resolution and more thematic detail for both the historic and the future land use/cover trajectory in Europe, compared to the current state-of-the-art land use forcing LUH2. The method embodies a novel approach of combining coarse-resolution land use transitions (from LUH2) with high-resolution land cover information (from ESA CCI LC; forest types) for generating time series of Plant Functional Types (PFT) maps tailored to the modelling community. The presented dataset/topic shows high significance and large potential to be used for further studies. It is unique, useful for future modelling experiments and presented in a complete, comprehensible form (accompanying the Reinhart et al. 2022 paper). Resulting datasets are accessible via the given identifier, accompanied with metadata, and a discussion of uncertainties in the article. Further, the results are compared to other recently published land use/cover change datasets, which places the findings in the current state of research. In addition, the paper is interestingly written and clearly structured.*

*Overall, the paper is well written and its content of wide scientific interest. I recommend publication after only a few minor, mostly technical revisions.*

Thank you very much for this positive feedback and the valuable comments and suggestions.

*Line 48: Why is a historic time series of 65 years required? Why is 1950 the starting year, is that a requirement from the RCMs?*

Yes, this is the requirement from the RCM community and in particular the EURO-CORDEX community. The historical simulations of the EURO-CORDEX experiments start in 1950. We added a short explanation in the introduction section.

*Line 100: "First, the ESA-CCI LC map for the year 2015, which has a native resolution of ~300 m globally, is aggregated to 0.1° resolution"*

*It was not clear to me how the aggregation was done. Which resampling method did you use, which classes/grid values did you include? I take notice that it is described in more detail in Reinhart et al. 2022, but a little more information would be good here."*

Thank you for your comment. In order to aggregate we used the SAGA GIS (Conrad et al. 2015) tool "Coverage of Categories". This tool calculates for each category (i.e. the ESA CCI LC class) the percentage it covers in each cell of the target grid system. We will add this information to the text.

*Lines 182-184: "Following the recommendations [..] natural vegetation (i.e., forest and shrubland) is only cleared and converted into grassland for land-use class transitions to pasture, while it remains unchanged for land-use class transitions from non-forested vegetation to rangeland."*

*What is the reason for this? Maybe the recommendations provide more justification. I am not sure if I understood it correctly: So, the dataset only includes transitions from natural vegetation to anthropogenic land use and no forest to shrubland or shrubland to natural grasslands? Perhaps you could provide some more justification and examples for this."*

Thank you for pointing this out. You are right; we did not explain this rule in detail. Ma et al. (2019) tested different transition rules for the conversion of natural vegetation (primary/secondary forest and primary/secondary non-forest) to managed land (i.e. cropland, rangeland, pasture). They compared the resulting land cover maps to available observation and found that rule 1, which we also apply, performs best. Ma et al. (2019) wrote: "Rule 1 (clearance of all vegetation for cropland and managed pasture, and only forest clearance for rangeland) is in fact the rule suggested in the

underlying HYDE dataset and its distinction between pasture and rangeland (Klein Goldewijk et al 2017).” And based on their results they wrote: "Therefore, recommendation of rule 1 over rule 2 is based on an assumption about the way in which rangeland versus managed pasture is established and managed which is also consistent with the recommendation in HYDE 3.2 dataset (Klein Goldewijk et al., 2017) that removes all vegetation when establishing cropland, urban land, or managed pasture, and leaves all vegetation when establishing rangeland, regardless of the underlying vegetation type.". We added more detail to the text.

The treatment of transition of natural land is explained in section 2.3: “Transitions from forest to non-forested vegetation (i.e., shrubland and grassland) and vice versa are not considered in the forward translation because these fields are zero in original LUH2 scenario data. Consequently, future afforestation and deforestation only occur if land use transitions related to land use classes urban, cropland, rangeland, and pasture are present. An exception is made for the three scenarios SSP1/RCP1.9, SSP1/RCP2.6, and SSP5/RCP4.5, where a separate dataset is provided for afforestation (Sect. 2.3.2).” Furthermore, we already discussed this aspect in detail in the “Indented use and limitations” section: “It needs to be noted, that the future land cover changes provided by LUCAS LUC consider anthropogenic land use changes, but do not account for potential latitudinal and altitudinal shifts of the of natural vegetation or in particular forest due to climate change (McDowell et al. 2020) because the underlying LUH2 data only provides land use changes due to anthropogenic activities. Therefore, the potential northwards expansion of forest in Europe, which is projected under different climate change scenarios (Dyderski et al. 2018), is not included in LUCAS LUC. Furthermore, in contrast to the historical LUCAS LUC reconstruction, the future forest composition does not change because the relative fractions of the tree and shrub PFTs stay constant during the forward translation. However, both the shift in the composition and the spatial distribution depend on the projected climate by the different ESMs/GCMs and are therefore uncertain.”

*Lines 282-232: "Treatment of irrigated cropland*

*Was there a reason to include irrigation as a land management option and not e.g. fertilizer application, pesticide usage, cropping frequency, crop types, etc.? It would be good to have more information/justification of why irrigation was selected as the only indicator of land management."*

Thank you for this comment. The LUCAS LUC dataset is tailored towards the needs of RCMs. Hence, we added PFTs that are used within RCMs. Irrigation can have strong impacts on the regional climate and is applied in some parts of Europe (e.g. Po Valley and parts of Spain). In the introduction, we wrote: “In addition, land management practices such as irrigation significantly alter local and regional climate (Lobell et al. 2009, Valmassoi et al. 2019) and should thus be accounted for in the reconstruction and scenarios.”

Therefore, a number of RCMs include a parameterization for irrigation. Other management practices are so far rarely implemented in RCMs (e.g. fertilizer) and associated processes are often not covered in current RCMs. For future RCM developments and applications, the LUCAS LUC PFT dataset can be extended such as for crop or forest management. We will expand our justification in the introduction and add a short outlook in the “Indented use and limitations” section, where we already wrote a discussion on irrigation.

*Tables 5 and 6: I do not quite understand the “forward in time” or “backward in time” from the captions in relation to the column and row labels in Table 5 and especially Table 6. I think that headers are needed to describe the “From…” column labels and the “To…” row labels, e.g. “PFT in time step 0” and “PFT in time step 1”. If Table 5 shows the “forward in time” transitions and Table 6 the “backward in time” transitions, why are there several entries in the “from URB” column in Table 5, whereas there are none in the “from URB” column of Table 6 (if it is read as backward in time, it*

*actually means transitions from something else to urban?). Also, the FOR-NFV transitions (no entries in Table 5 and 2 entries in Table 6) are not clear to me.*

Thank you for this valuable comment and for your suggestions. The column and row labels refer to the land use transitions provide by LUH2. Hence, they are the same both for the forward and backward translation. However, the changes provided in the cells of the table have a different meaning for the backward and forward transitions. As you pointed out, the changes in PFT fraction are from timestep t to timestep t+1 in the forward translation and from timestep t to timestep t-1 in the backward translation. In order to clarify this, we included this explanation in the table captions.

You are right, the "from URB" column is empty in the backward translation. This transition would indicated a historical deurbanization, which rarely happens and, in addition, these changes are zero in LUH2. Hence, we already wrote: "Since the historical transitions from urban to any other LUH2 land use class are zero, these transitions are not considered." In order to clarify this, we will add: "and are therefore not listed in Table 6."

*Lines 571-572: It is not surprising that LUCAS LUC land cover changes are similar to LUH2, as the LUH2 is a main input for generating the dataset. I suggest adding in the introduction why LUH2 is so important for the methodology and in the end for the emergence of LUCAS LUC (possibly because there are no other annual land use change datasets with future scenarios).*

With this sentence we wanted to state that we are indeed following the LUH2 changes. The main reason to use the LUH2 data for LUCAS LUC was that we aimed for a dataset that is in line with the forcings from the CMIP6 experiment, where LUH2 is used as a land surface forcing for the ESMs/GCMs. These simulations will be used for the EURO-CORDEX downscaling experiments within the CORDEX phase 2. In the introduction, we wrote that this is a requirement for the new dataset: "LULCC forcing should necessarily follow the overall trends employed by the driving Global Climate Models/Earth System Models (GCM/ESM) to be consistent with the boundary forcing as it is done for other forcing data such as greenhouse gas concentrations or aerosol emissions (Taranu et al. 2022, Wohland 2022).". Now we mention in the introduction that we developed an LUT approach that generates a new land cover input dataset for RCMs, which follows the land use changes provided by LUH2.

*Line 72: Word duplication, please remove one "keeping".*

Thank you. We removed it.

*Line 289: "I think "Tsendbazar et al. (2021)" is not the right reference for the Copernicus LC100 dataset and brackets are missing. Please put the correct dataset reference here:*

*Buchhorn, M., Smets, B., Bertels, L., Lesiv, M., Tsendbazar, N.E., Herold, M., Fritz, S., 2020. Copernicus Global Land Service: Land Cover 100m: collection 3: epoch 2015-2019: Globe. Version V3. 0.1.*

*https://doi.org/10.5281/zenodo.3939038; https://doi.org/10.5281/zenodo.3518026; https://doi.org/10.5281/zenodo.3518036; https://doi.org/10.5281/zenodo.3518038, https://doi.org/10.5281/zenodo.3939050"*

Thank you for detecting this error. We added the correct reference to the text and reference section.

All suggested technical corrections were implemented:

*Line 386: Remove „in" from „and in especially HILDA+".*

*Line 399: "lager" should be "larger".*

*Line 404: "HIDLA+" should be "HILDA+".*

*Figure 6, caption: "show" should be "shown".*

*Line 471: A comma is missing after "While grassland cover strongly increases in one scenario (Fig. 13b) in the IP region".*

*Line 494: "a increase" should be "an increase".*

*Line 544: Commas are missing before and after "averaged over Europe".*

*Line 608: "a extreme" should be "an extreme".*

*Line 664: Add an "and" after "The LUCAS LUC datasets were already produced for other CORDEX regions (Hoffmann et al., 2021)".*

---

## Author Response (AR1)

Reviewer 1

| Line | Comment | Answer | Work in Text | Line |
|------|---------|--------|--------------|------|
| 72 | delete extra "keeping" | done | removed | 72 |
| 399 | "lager" should be "larger" | done | added | 401 |
| 546 | delete the first "based" | done | removed | 548 |
| 609 | "emphasize" should be "emphasizes" | done | added | 614 |
| 621 | "giving" should be "given" | The change needs to be made in L612, not L621 | changed | 617 |
| 619 | "changes" should be "change" | done | removed | 623 |
| 637 | "documentation" should be "document" | done | changed | 642 |
| 556 | "currently validated" should be "currently being validated" | The change needs to be made in L665, not L556 | added | 670 |

Reviewer 2

| Line | Comment | Answer | Work in Text | Line |
|---|---|---|---|---|
| 48 | Why is a historic time series of 65 years required? Why is 1950 the starting year, is that a requirement from the RCMs? | Yes, this is the requirement from the RCM community and in particular the EURO-CORDEX community. The historical simulations of the EURO-CORDEX experiments start in 1950. We added a short explanation in the introduction section. | In the next phases of LUCAS and within EURO-CORDEX (Jacob et al., 2020), it is planned to conduct simulations with past and future LULCC forcing at a ~12.5 km (i.e., EUR-11 domain) horizontal resolution. For some specific sub-regions in Europe, simulations will be also carried out at convection permitting resolutions. This approach implies new requirements for LULCC reconstructions and scenarios: 1) A high spatial resolution (1 km or below) over an extent that covers the entire EURO-CORDEX domain in order to enable the investigation of LULCC impacts on small-scale processes such as local wind systems, convection, boundary layer processes, and scale-interactions (Mahmood et al., 2014). 2) A temporal coverage starting from 1950, which is the time frame defined in the EURO-CORDEX historical experiments. Further, the LULCC product should extend until 2100 for analyzing the impact of several Shared Socioeconomic Pathways (SSPs) and Representative Concentration Pathways (RCPs) scenarios accounting for both changes to anthropogenic emissions and LULCC. | 40-50 |
| 100 | "First, the ESA-CCI LC map for the year 2015, which has a native resolution of ~300 m globally, is aggregated to 0.1 ◦ resolution" It was not clear to me how the aggregation was done. Which resampling method did you use, which classes/grid values did you include? I take notice that it is described in more detail in Reinhart et al. 2022, but a little more information would be good here. | Thank you for your comment. In order to aggregate we used the SAGA GIS (Conrad et al. 2015) tool "Coverage of Categories". This tool calculates for each category (i.e. the ESA CCI LC class) the percentage it covers in each cell of the target grid system. We will add this information to the text. | First, the ESA-CCI LC map for the year 2015, which has a native resolution of ~300 m globally, is aggregated to 0.1° resolution using the SAGA GIS (Conrad et al., 2015) tool Coverage of Categories. It computes the percentage of each ESA-CCI LC class for the 0.1x0.1° grid cells. | 100-103 |
| 182-184 | "Following the recommendations [..] natural vegetation (i.e., forest and shrubland) is only cleared and converted into grassland for land-use class transitions to pasture, while it remains unchanged for land-use class transitions from non-forested vegetation to rangeland." What is the reason for this? Maybe the recommendations provide more justification. I am not sure if I understood it correctly: So, the dataset only includes transitions from natural vegetation to anthropogenic land use and no forest to shrubland or shrubland to natural grasslands? Perhaps you | Thank you for pointing this out. You are right; we did not explain this rule in detail. Ma et al. (2019) tested different transition rules for the conversion of natural vegetation (primary/secondary forest and primary/secondary non-forest) to managed land (i.e. cropland, rangeland, pasture). They compared the resulting land cover maps to available observation and found that rule 1, which we also apply, performs best. Ma et al. (2019) wrote: "Rule 1 (clearance of all vegetation for cropland and managed pasture, and only forest clearance for rangeland) is in fact the rule suggested in the underlying HYDE dataset and its distinction between pasture and rangeland (Klein Goldewijk et al 2017)." And based on their results they wrote: "Therefore, recommendation of rule 1 over rule 2 is based on an assumption about the way in which rangeland versus managed pasture is established and managed which is also consistent with the recommendation in HYDE 3.2 dataset | The transition rules are defined to ensure that the changes in cropland are as close to the LUH2 changes as possible. In contrast to other LUTs, urban transitions are included. Following the recommendations by Ma et al. (2020) and Hurtt et al. (2020), natural vegetation (i.e., forest and shrubland) is cleared and converted into grassland only for land use transitions to pasture, while it remains unchanged for land use transitions from non-forested vegetation to rangeland. Hence, it is assumed that vegetation is cleared if the land is converted into managed pasture while it remains unchanged if rangeland is established. An exception to this general rule is the transition from forest to rangeland when the land will be used for livestock grazing. | 184-187 |

| | could provide some more justification and examples for this. | (Klein Goldewijk et al., 2017) that removes all vegetation when establishing cropland, urban land, or managed pasture, and leaves all vegetation when establishing rangeland, regardless of the underlying vegetation type.". We added more detail to the text.

The treatment of transition of natural land is explained in section 2.3: "Transitions from forest to non-forested vegetation (i.e., shrubland and grassland) and vice versa are not considered in the forward translation because these fields are zero in original LUH2 scenario data. Consequently, future afforestation and deforestation only occur if land use transitions related to land use classes urban, cropland, rangeland, and pasture are present. An exception is made for the three scenarios SSP1/RCP1.9, SSP1/RCP2.6, and SSP5/RCP4.5, where a separate dataset is provided for afforestation (Sect. 2.3.2)." Furthermore, we already discussed this aspect in detail in the "Indented use and limitations" section: "It needs to be noted, that the future land cover changes provided by LUCAS LUC consider anthropogenic land use changes, but do not account for potential latitudinal and altitudinal shifts of the of natural vegetation or in particular forest due to climate change (McDowell et al. 2020) because the underlying LUH2 data only provides land use changes due to anthropogenic activities. Therefore, the potential northwards expansion of forest in Europe, which is projected under different climate change scenarios (Dyderski et al. 2018), is not included in LUCAS LUC. Furthermore, in contrast to the historical LUCAS LUC reconstruction, the future forest composition does not change because the relative fractions of the tree and shrub PFTs stay constant during the forward translation. However, both the shift in the composition and the spatial distribution depend on the projected climate by the different ESMs/GCMs and are therefore uncertain." | | |

| 282-232 | **Treatment of irrigated cropland** Was there a reason to include irrigation as a land management option and not e.g. fertilizer application, pesticide usage, cropping frequency, crop types, etc.? It would be good to have more information/justification of why irrigation was selected as the only indicator of land management. | Thank you for this comment. The LUCAS LUC dataset is tailored towards the needs of RCMs. Hence, we added PFTs that are used within RCMs. Irrigation can have strong impacts on the regional climate and is applied in some parts of Europe (e.g. Po Valley and parts of Spain). In the introduction, we wrote: "In addition, land management practices such as irrigation significantly alter local and regi+D5+C5 | 4) A choice of land use and land cover classes that matches the specific needs of current RCMs. For instance, at scales of ~50 km and lower, urban land cover plays an important role (Chapman et al., 2019; Daniel et al., 2019; Katzfey et al., 2020) and should be represented. Moreover, at these scales the ratio of needleleaf to broadleaf trees becomes a meaningful aspect to consider (Naudts et al., 2016; Schwaab et al., 2020). Finally, land management practices such as irrigation significantly alter local and regional climate and are implemented in RCMs (Lobell et al., 2009; Valmassoi et al., 2019; Asmus et al., submitted). Thus, irrigation changes should be accounted for in the reconstruction and scenarios. | 54-59 |
| | | | While a number of RCMs include a parameterization for irrigation. Other management practices are so far rarely implemented in RCMs (e.g. fertilizer) and associated processes are often not covered in current RCMs. For future RCM developments and applications, the LUCAS LUC PFT dataset can be extended such as for crop or forest management. | 603-606 |
| Tables 5 and 6 | I do not quite understand the "forward in time" or "backward in time" from the captions in relation to the column and row labels in Table 5 and especially Table 6. I think that headers are needed to describe the "From…" column labels and the "To…" row labels, e.g. "PFT in time step 0" and "PFT in time step 1". If Table 5 shows the "forward in time" transitions and Table 6 the "backward in time" transitions, why are there several entries in the "from URB" column in Table 5, whereas there are none in the "from URB" column of Table 6 (if it is read as backward in time, it actually means transitions from something else to urban?). Also, the FOR-NFV transitions (no entries in Table 5 and 2 entries in Table 6) are not clear to me. | Thank you for this valuable comment and for your suggestions. The column and row labels refer to the land use transitions provide by LUH2. Hence, they are the same both for the forward and backward translation. However, the changes provided in the cells of the table have a different meaning for the backward and forward transitions. As you pointed out, the changes in PFT fraction are from timestep t to timestep t+1 in the forward translation and from timestep t to timestep t-1 in the backward translation. In order to clarify this, we included this explanation in the table captions.

You are right, the "from URB" column is empty in the backward translation. This transition would indicated a historical deurbanization, which rarely happens and, in addition, these changes are zero in LUH2. Hence, we already wrote: "Since the historical transitions from urban to any other LUH2 land use class are zero, these transitions are not considered." In order to clarify this, we will add: "and are therefore not listed in Table 6." | LUT rules for the translation of LUT class changes into PFT changes forward in time using the PFT group definitions given in table 2. This means the transitions refer to the changes in PFT fraction from timestep t to timestep t+1.

Please note that the transitions provided by LUH2 are the same as in table 2 but the changes in PFTs given in this table are imposed backward in time. This means the transitions refer to the changes in PFT fraction from timestep t to timestep t-1.

Since the historical transitions from urban to any other LUH2 land use class are zero, these are not listed in table 6. | Table 5

Table 6

196-197 |

| 571-572 | It is not surprising that LUCAS LUC land cover changes are similar to LUH2, as the LUH2 is a main input for generating the dataset. I suggest adding in the introduction why LUH2 is so important for the methodology and in the end for the emergence of LUCAS LUC (possibly because there are no other annual land use change datasets with future scenarios). | With this sentence we wanted to state that we are indeed following the LUH2 changes. The main reason to use the LUH2 data for LUCAS LUC was that we aimed for a dataset that is in line with the forcings from the CMIP6 experiment, where LUH2 is used as a land surface forcing for the ESMs/GCMs. These simulations will be used for the EURO-CORDEX downscaling experiments within the CORDEX phase 2. In the introduction, we wrote that this is a requirement for the new dataset: "LULCC forcing should necessarily follow the overall trends employed by the driving Global Climate Models/Earth System Models (GCM/ESM) to be consistent with the boundary forcing as it is done for other forcing data such as greenhouse gas concentrations or aerosol emissions (Taranu et al. 2022, Wohland 2022).". Now, we mention in the introduction that we developed an LUT approach that generates a new land cover input dataset for RCMs, which follows the land use changes provided by LUH2. | 3) A LULCC forcing generally consistent with the LULCC forcing employed by the driving Global Climate Models/Earth System Models (GCM/ESM) as it is the case for other forcing data such as greenhouse gas or aerosol emissions (Taranu et al., 2022; Wohland, 2022).Consequently, we developed a new LUT approach, which also accounts for changes in the distribution of natural vegetation types and urban areas, and generated a new land cover input dataset for RCMs consistent with LUH2, which is also used in CMIP6. | 51-5377-78 |

| Technichal corrections | | | | |
|---|---|---|---|---|
| 72 | Word duplication, please remove one "keeping". | done | removed | 72 |
| 289 | I think "Tsendbazar et al. (2021)" is not the right reference for the Copernicus LC100 dataset and brackets are missing. Please put the correct dataset reference here:Buchhorn, M., Smets, B., Bertels, L., Lesiv, M., Tsendbazar, N.E., Herold, M., Fritz, S., 2020. Copernicus Global Land Service: Land Cover 100m: collection 3: epoch 2015-2019: Globe. Version V3. 0.1.https://doi.org/10.5281/zenodo.3939038; https://doi.org/10.5281/zenodo.3518026; https://doi.org/10.5281/zenodo.3518036; https://doi.org/10.5281/zenodo.3518038, https://doi.org/10.5281/zenodo.3939050 | Thank you for detecting this error. We added the correct reference to the text and reference section. | references exchanged in text and bib file | 291 |
| 386 | Remove „in" from „and in especially HILDA+". | done | removed | 388 |
| 399 | "lager" should be "larger". | done | added | 401 |
| 404 | "HIDLA+" should be "HILDA+". | done | changed | 406 |
| Figure 6, caption | "show" should be "shown". | done | added | Figure 6, caption |
| 471 | A comma is missing after "While grassland cover strongly increases in one scenario (Fig. 13b) in the IP region". | done | added | 473 |
| 494 | "a increase" should be "an increase". | done | added | 496 |
| 544 | Commas are missing before and after "averaged over Europe". | done | added | 546 |
| 608 | "a extreme" should be "an extreme". | done | added | 613 |
| 664 | Add an "and" after "The LUCAS LUC datasets were already produced for other CORDEX regions (Hoffmann et al., 2021)". | done | added | 670 |

---

## Author Response (AR2)

Dear Dr Carlson,

Thank you for your comments! I have changed the text accordingly.

Small changes, subject to authors' discretion. These changes can all occur at proof stage.

Line 76,77: Resolve whether "also used for CMIP6" refers to LUH2 or to this new "land cover input dataset". Most readers will know answer but language as used allows misinterpretation.

*Consequently, we developed a new LUT approach, which also accounts for changes in the distribution of natural vegetation types and urban areas, and generated a new land cover input dataset for RCMs that is consistent with the LUH2 dataset, which is also used in CMIP6.*

Line 112: CRU is 'geographically' "out of range" or in terms of temperature terms? Former = yes. Latter I don't know without checking. Again, language improvements will clarify and assist for general readers.

*The HLZs for the European domain are computed from atmospheric observations of temperature and precipitation taken from the E-OBS dataset and the CRU dataset (outside the geographical range of E-OBS range).*

Line 673: Title "Appendix A" should appear here of at head of appendix tables on page 47?

*Thank you! It is fixed now.*

Thanks authors for patience and persistence!

Best regards

Peter Hoffmann